# *Tlr9* deficiency in B cells leads to obesity by promoting inflammation and gut dysbiosis

Pai Wang ®[1,2,7], Xin Yang[2,3,7], Luyao Zhang[1,2], Sha Sha[2,4], Juan Huang[2], Jian Peng[2], Jianlei Gu[5], James Alexander Pearson ®[2,6], Youjia Hu[2], Hongyu Zhao ®[5], F. Susan Wong[6], Quan Wang[1]✉ & Li Wen ®[2]✉

Toll-like receptor 9 (TLR9) recognizes bacterial, viral and self DNA and play an important role in immunity and inflammation. However, the role of TLR9 in obesity is less well-studied. Here, we generate B-cell-specific *Tlr9*-deficient (*Tlr9fl/fl/Cd19Cre+/-*, KO) B6 mice and model obesity using a high-fat diet. Compared with control mice, B-cell-specific-*Tlr9*-deficient mice exhibited increased fat tissue inflammation, weight gain, and impaired glucose and insulin tolerance. Furthermore, the frequencies of IL-10-producing-B cells and marginal zone B cells were reduced, and those of follicular and germinal center B cells were increased. This was associated with increased frequencies of IFNγ-producing-T cells and increased follicular helper cells. In addition, gut microbiota from the KO mice induced a pro-inflammatory state leading to immunological and metabolic dysregulation when transferred to germ-free mice. Using 16 S rRNA gene sequencing, we identify altered gut microbial communities including reduced Lachnospiraceae, which may play a role in altered metabolism in KO mice. We identify an important network involving *Tlr9*, *Irf4* and *Il-10* interconnecting metabolic homeostasis, with the function of B and T cells, and gut microbiota in obesity.

Obesity has become a major public health burden. There are currently over 1.9 billion people, worldwide who are overweight or obese, leading to a sharp rise in related health complications[1], including cardiovascular disease, liver disease and cancer[2,3]. Obesity leads to chronic inflammation that involves both innate and adaptive immunity[4–10].

Toll-like receptors (TLRs) are pattern recognition receptors (PRRs) that play a central role in the development and function of the immune system[11]. Innate immune cells including B cells express various PRRs that recognize pathogen-associated molecular patterns (PAMPs) derived from various microbes and induce inflammatory immune responses[12,13]. TLR9 recognizes unmethylated CpG DNA from bacteria

and initiates type I interferon (IFN) responses by immune cells[14]. TLR9 is expressed on a large range of immune cells, including B cells[15–17], T cells[18,19], macrophages[19,20], dendritic cells[17,21] and NK cells[22,23] but is particularly highly expressed in B cells and dendritic cells. TLR9 may play an important role in development of type 2 diabetes (T2D), as the expression of TLR9 is associated with immune cell infiltration in adipose tissue[24]. These include positive associations between TLR9 and memory B cells, and negative associations between TLR9 and naïve B cells[25]. TLR9 ligands are important for B cell activation and antibody responses to microbial antigens, vaccination, and metabolic diseases[26–30]. C57BL/6 mice with total body knockout of TLR9 exhibited higher body weight and fat content on a high-fat diet and the

---

[1]Department of Gastrocolorectal Surgery, General Surgery Center, The First Hospital of Jilin University, Changchun, Jilin, China. [2]Section of Endocrinology, Department of Internal Medicine, School of Medicine, Yale University, New Haven, CT, USA. [3]Department of Food Science and Technology, School of Agriculture and Biology, Shanghai Jiao Tong University, Shanghai, China. [4]Department of Nephrology, The First Affiliated Hospital of Shandong First Medical University, Jinan, Shandong, China. [5]Department of Biostatistics, Yale School of Public Health, New Haven, CT, USA. [6]Division of Infection and Immunity, School of Medicine and Systems Immunity University Research Institute, Cardiff University, Cardiff, UK. [7]These authors contributed equally: Pai Wang, Xin Yang. ✉e-mail: wquan@jlu.edu.cn; li.wen@yale.edu

underlying mechanism may be associated with a significant increase in M1 macrophages and Th1 cells[31].

TLR9 signaling can modulate the composition and function of gut microbiota[32–34]. B cells also shape the composition of gut microbiota directly through production of antimicrobial peptides and other factors that help to control the growth of specific bacterial species in the gut[35,36]. Recent studies suggest that TLR9 plays an important role in obesity, which is related to tissue inflammation and insulin resistance[37,38]. Crosstalk between B cells and gut microbiota governs immunity and metabolism in the gut, which could lead to metabolic syndrome including obesity and insulin resistance[39,40]. However, the role that TLR9 plays in B cells in the context of obesity, and how TLR9 in B cells affects adaptive immunity and gut microbiota in obesity is unclear. Using total body *Tlr9* knock-out mice in our previous study, we observed that TLR9 plays a role in glucose tolerance regardless of the genetic background[41]. Furthermore, in the NOD mouse model of type 1 diabetes, *Tlr9* deficiency in B cells promoted immune tolerance through IL-10[42]. Here, to investigate the intrinsic role of B cell-TLR9 in obesity, we generate C57BL/6 (B6) mice with B cell-specific deficiency of *Tlr9* (*Tlr9*^fl/fl^/*Cd19Cre*^+/-^ B6) and use a high fat diet-induced obesity (HFDIO) model system to study the effect of B cell-specific deficiency of *Tlr9* on local and systemic immunity and the effect on gut microbiota. Our study also provides insight into the impact of TLR9 in B cells on obesity, associated with immunological, metabolic, and gut microbiological abnormalities.

## Results

### B cell-specific Tlr9 deficiency leads to obesity

To investigate the role of B cell-TLR9 in obesity, we generated B cell-specific *Tlr9*-deficient C57BL/6 mice (*Tlr9*^fl/fl^/*Cd19Cre*^+/-^B6), and studied both *Tlr9*^fl/fl^/*Cd19Cre*^+/-^ B6 mice (designated as KO group) and *Tlr9*^fl/fl^/*Cd19Cre*^-/-^ B6 littermate controls (designated as Ctr group). It is noteworthy that *Cd19Cre* is a knockin mutant, and homozygous *Cd19Cre* (*Cd19Cre*^+/+^) mice have impaired B cell development; however, B cell development in *Cd19Cre* heterozygous (*Cd19Cre*^+/-^) mice was comparable to *Tlr9*^fl/fl^ mice, wild type for *Cd19Cre* (Fig. S1). We further confirmed the phenotype of heterozygotes on both normal and high fat diets. To verify if *Cd19Cre*^+/-^ B6 control mice express the same phenotype as the *Tlr9*^fl/fl^/*Cd19Cre*^-/-^ B6 control mice, we fed three groups of mice - *Tlr9*^fl/fl^ B6 (Ctr_flox), *Cd19Cre*^+/-^ B6 (Ctr_Cre), and *Tlr9*^fl/fl^ *Cd19Cre*^+/-^ B6 (KO) with a high-fat diet for 6 weeks (Fig. S2). We observed no significant difference in body weight between the two control groups, Ctr_flox and Ctr_Cre, while the KO group had significantly higher body weight compared to mice in Ctr_flox and Ctr_Cre (Fig. S2A). Additionally, we found no significant differences in intraperitoneal glucose tolerance tests (IPGTT) between the two control groups. Thus, *Tlr9*^fl/fl^/*Cd19Cre*^+/-^ B6 (KO) mice and *Tlr9*^fl/fl^/*Cd19Cre*^-/-^ B6 (Ctr_flox) control mice were used in the following experiments. We investigated male mice which were fed with a high fat diet at 6-weeks of age for 12 weeks (Fig. 1A), monitoring body weight weekly and observed that the KO group gained significantly more weight than their control counterparts (Fig. 1B, C). The KO group also had higher

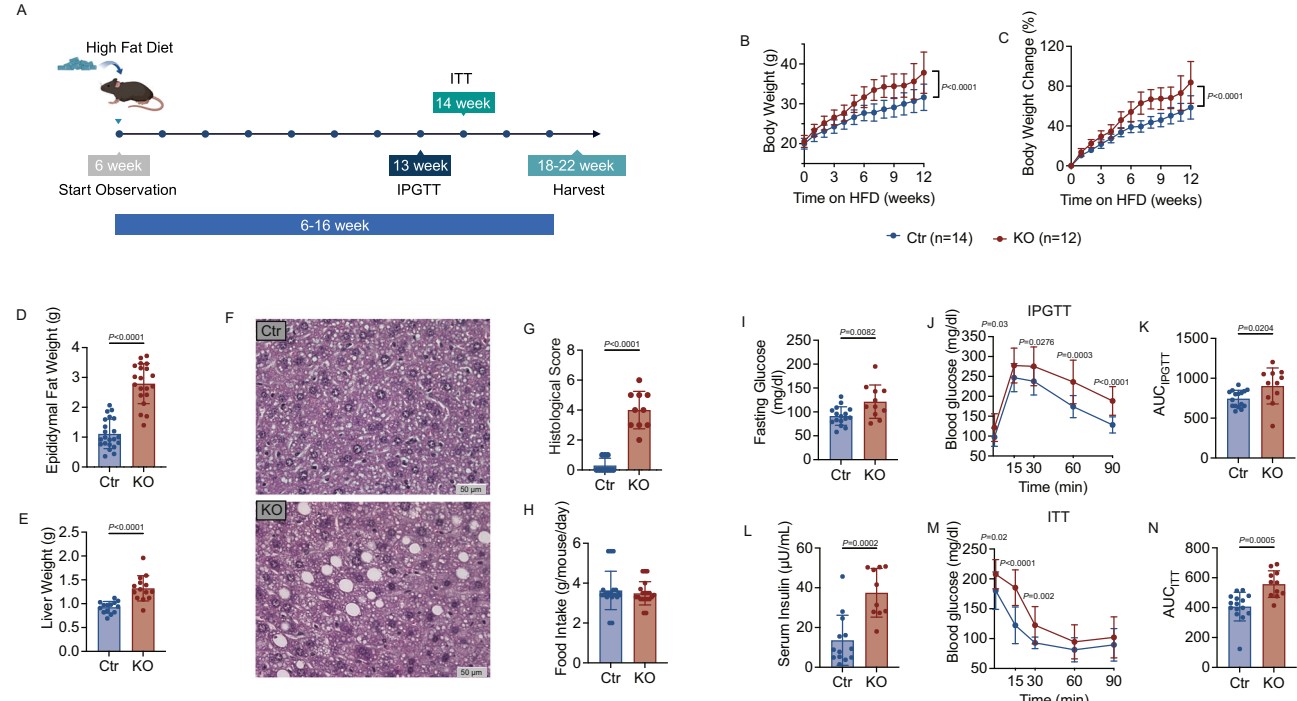

**Fig. 1 | B Cell–Specific *Tlr9* deficiency accelerates Obesity.** *Tlr9* deletion in B cells (*Tlr9*^fl/fl^/*Cd19Cre*^+/-^ B6) promotes high fat diet-induced obesity (HFDIO) in C57BL/6 mice. *Tlr9*^fl/fl^/*Cd19Cre*^+/-^ B6 (KO) male mice and *Tlr9*^fl/fl^/*Cd19Cre*^-/-^ B6 control (Ctr) male mice were fed with HFD at 6 weeks of age. **A** Experimental design of high fat diet induced obesity (HFDIO). The mice were monitored body weight (BW) weekly at the beginning of HFD (6 weeks of age). Intraperitoneal glucose tolerance test (IPGTT) was assessed at 13 weeks old. Insulin tolerance test (ITT) was assessed at 14 weeks old. The mice were terminated at 18–22 weeks of age. Graph was created with BioRender.com. **B** Net body weight change: body weight was measured weekly from the commencement of HFD (week 0) (KO: *n* = 12 mice, Ctr: *n* = 14 mice). **C** Body weight change (%) of KO and Ctr mice. **D** Epididymal fat weight of the KO and Ctr mice after 12 weeks of HFD (KO: *n* = 20 mice, Ctr: *n* = 22 mice). **E** Liver weight of the KO mice and the control mice after 12 weeks of HFD (*n* = 14 mice per group).

**F** Representative liver sections after staining with H&E. 40×. **G** Histological scores of liver sections with H&E. **H** Food intake of the KO and Ctr mice (*n* = 20 mice per group). **I** Fasting glucose levels of the KO and Ctr mice (KO: *n* = 11 mice, Ctr: *n* = 16 mice). **J** IPGTT results of the KO and the control mice at 13 weeks old (KO: n = 11 mice, Ctr: *n* = 16 mice). **K** Area under the curve (AUC) of the IPGTT of **J**. **L** Serum insulin levels of the KO and Ctr mice (KO: *n* = 10 mice, Ctr: *n* = 13 mice). **M** ITT results of the KO and control mice at 14 weeks old (KO: *n* = 11 mice, Ctr: *n* = 15 mice). **N** Area under the curve (AUC) of the ITT of **M**. All the data were from two independent experiments. **A**, **B** were analyzed using two-way ANOVA and **C**–**M** were analyzed using two-tailed Student's *t*-test. **I**, **L** were analyzed using multiple Student's *t*-test. The data are presented as mean ± SD. The experiment was performed 3 times and the pooled results are presented in the figure.

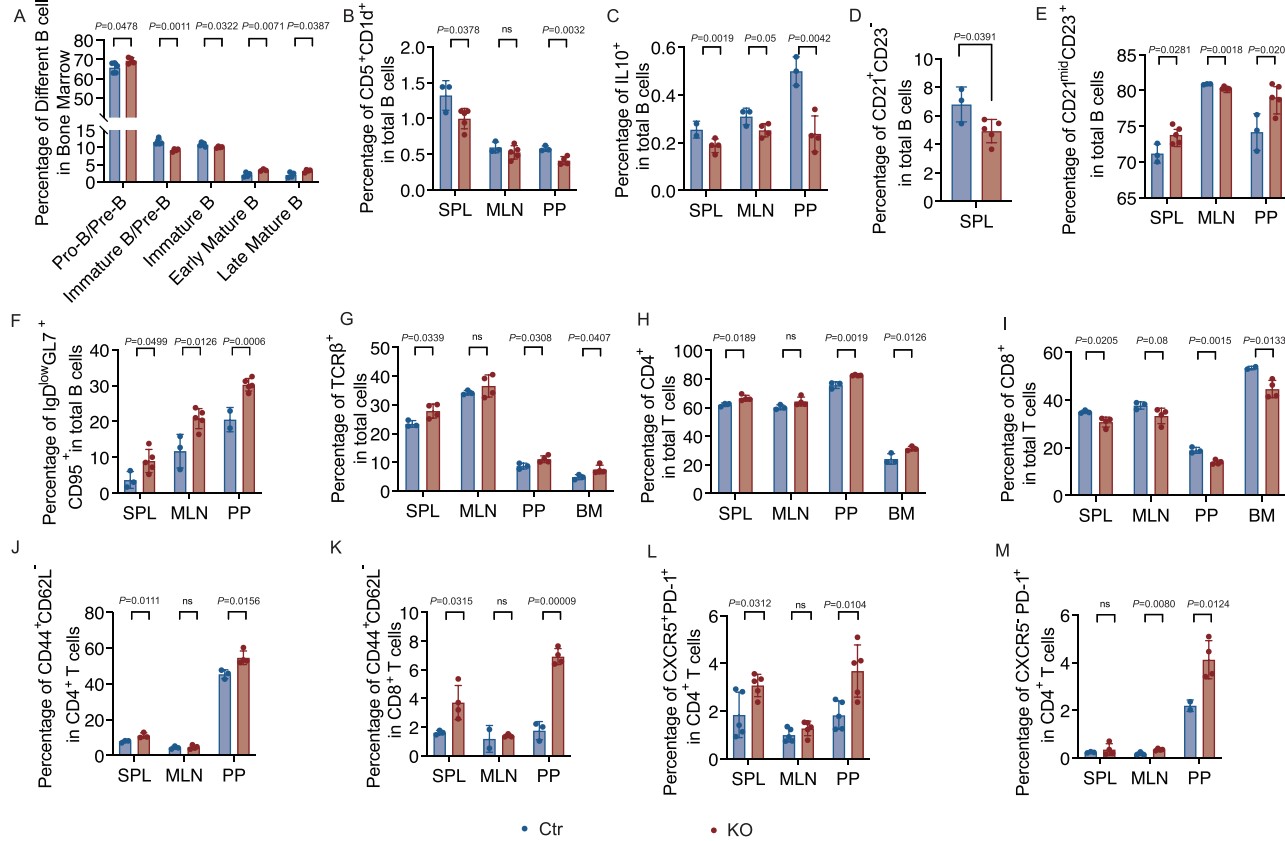

**Fig. 2 | *Tlr9* deficiency in B cells alters the phenotype of different immune cell subsets.** Immune cells from different lymphoid tissues (as indicated) in *Tlr9fl/fl/Cd19Cre+/-* B6 (KO) and *Tlr9fl/fl/Cd19Cre-/-* B6 (Ctr) mice (all males, 18–22-week-old) were prepared and analyzed using flow cytometry after staining with different markers and Zombie Dye. All the samples were first gated on Zombie Dye negative cells followed by further gating on different immune cell subsets. **A**–**F** Immunephenotype of B cells (gated on B220+) in KO and Ctr mice. **A** B cells were first gated on B220+ cells. The proportion of Pro-Pre B cells (CD24+,CD43-, IgD-, IgM-), immature-Pre B cells (CD24+,CD43-, IgD-, IgMmid), immature B cells (CD24+,CD43-, IgD-, IgM+), early mature B cells (CD24+,CD43-, IgD+, IgM+) and late mature B cells (CD24+,CD43-, IgD+,IgM-), in the bone marrow (KO: *n* = 4 mice, Ctr: *n* = 5 mice). **B** The proportion of CD5+CD1d+ B cells in SPL, MLN, PP of the KO and the Ctr mice (KO: *n* = 5 mice, Ctr: *n* = 3 mice). **C** The proportion of B cells expressing IL-10 (KO: *n* = 4 mice s, Ctr: *n* = 3 mice). **D** The proportion of CD21+CD23- marginal zone (MZ) B cells (KO: n = 5 mice, Ctr: *n* = 3 mice). **E** The proportion of CD21midCD23+ follicular

(FO) B cells (KO: *n* = 5 mice, Ctr: *n* = 3 mice). **F** The proportion of IgDlowGL7+CD95+ germinal center (GC) B cells (KO: *n* = 5 mice, Ctr: *n* = 3 mice). **G**–**M** Immunephenotype of T cells (gated on TCRβ+) in KO mice and Ctr mice by flow cytometric analysis. **G** The proportion of total TCRαβ+ T cells (KO: *n* = 4 mice, Ctr: *n* = 3 mice). **H** Frequency of CD4+ T cells (KO: *n* = 4 mice, Ctr: *n* = 3 mice). **I** The proportion of CD8+ T cells (KO: *n* = 4 mice, Ctr: *n* = 3 mice). **J** The proportion of CD44+CD62L- memory CD4+ T cells (KO: *n* = 4 mice, Ctr: *n* = 3 mice). **K** The proportion of CD44+CD62L- memory CD8+ T cells (KO: *n* = 4 mice, Ctr: *n* = 3 mice). **L** The proportion of CXCR5+ PD-1+ follicular T helper (TFH) CD4+ T cells (KO: *n* = 5 mice, Ctr: *n* = 5 mice). **M** The proportion of CXCR5-PD-1+ peripheral helper (TPH) CD4+ T cells (KO: *n* = 4 mice, Ctr: *n* = 3 mice). All the experiments were repeated three times with similar results. Data were analyzed using two-tailed Student's *t*-test. The data are presented as mean ± SD. The data presented in the figures are from one of the experiments. The total cell numbers of the immune cells in each subset is presented in Supplementary Fig. 3.

epididymal fat-pad and liver weights (Fig. 1D, E), and there was more fat accumulation in the liver (Fig. 1F, G). However, food intake was not significantly different between the two groups (Fig. 1H). Next, we measured the fasting glucose and insulin levels and performed glucose and insulin tolerance tests in the two groups of mice. We observed a higher fasting glucose level (Fig. 1I) in the KO group, and the KO group had impaired glucose tolerance (Fig. 1J), measured as a higher area under the curve (AUC) of the IPGTT, compared with the Ctr mice (Fig. 1K). The KO group also had higher fasting serum insulin (Fig. 1L), and reduced insulin sensitivity as shown by higher glucose levels in response to insulin injection (Fig. 1M), as well as higher AUC in the ITT (Fig. 1N). These data suggested that TLR9 in B cells plays a protective role in obesity development, as B cells without TLR9 accelerated obesity and related metabolic abnormalities.

## Tlr9 deficiency in B cells alters the phenotype of different immune cells

To investigate the intrinsic role of *Tlr9* in B cell development and differentiation, we evaluated B cells and other types of immune cells in

central and peripheral lymphoid tissues. In the bone marrow (BM), we found elevated pro/pre-B cells (CD24+, CD43-, IgD-, IgM-), early mature B cells (CD24+CD43-IgD+IgM+) and late mature B cells (CD24+CD43-IgD+IgM-), but reduced immature B cells (CD24+CD43-IgD-IgM+) in the KO group (Fig. 2A and Fig. S3A). The peripheral B cell subsets were also altered, as shown by a significant decrease in CD1d+CD5+ B cells, which are known to contain IL-10-producing B regulatory cells[43,44], in spleen (SPL), mesenteric lymph nodes (MLN) and Peyer's patches (PP), from KO group compared with Ctr group (Fig. 2B and Fig. S3B). Next, we assessed IL-10-producing B cells in different lymphoid tissues and found that they were significantly decreased in the PP of KO group when compared with the Ctr group (Fig. 2C and Fig. S3C). Phenotypic analysis also revealed a significant decrease in marginal zone (MZ) B cells (CD21+CD23-) in SPL of the KO group compared with the Ctr group (Fig. 2D and Fig. S3D); whereas follicular (FO) B cells (CD21midCD23+) were also significantly increased in SPL of the KO group compared with the Ctr group (Fig. 2E) but the total number of splenic FO B cells was decreased (Fig. S3E). In addition, increased germinal center (GC) B cells (IgDlowGL7+CD95+)

were found in SPL, MLNs, and PPs (Fig. 2F and Fig. S3F). Taken together, our data suggest that *Tlr9* deficiency in B cells modulated the differentiation of BM B cells and altered the proportion and the cell number of different B cell subsets in the peripheral lymphoid tissues.

In addition to the impact on B cells, we discovered that *Tlr9*-deficient B cells also affected T cells. The proportion and total T cell number (TCRβ⁺) were increased in most, if not all, the lymphoid tissues examined in KO group compared with the Ctr group (Fig. 2G and Fig. S3G). CD4⁺ T cells were significantly increased in the lymphoid tissue examined in KO group compared with the Ctrl group (Fig. 2H and Fig. S3H), whereas the proportion of CD8⁺ T cells was significantly decreased (Fig. 2I), but the total number of the CD8⁺ T cells was increased in the KO group compared with the Ctrl group (Fig. S3I). Furthermore, there were more CD4⁺ and CD8⁺ memory T cells (CD44⁺CD62L⁻) in the KO group (Fig. 2J, K and Fig. S3J, K). Interestingly, phenotypic analysis revealed a significant increase in T follicular helper (T_FH) cells (CXCR5⁺PD-1⁺CD4⁺) and T peripheral helper (T_PH) cells (CXCR5⁻ PD-1⁺CD4⁺) in the KO group compared with the Ctr group (Fig. 2L, M and Fig. S3L, M), both of which are closely related to B cell differentiation and function[45–47]. There was a similar pattern in various subsets of immune cells, especially the B cell phenotype, in normal food-fed KO mice compared to the Ctr group (Fig. S4 and Fig. S5). Our results demonstrated that *Tlr9* deficiency in B cells not only affected the differentiation of B cells but also affected T cells, which may in turn influence B cells.

### Tlr9 deficiency in B cells alters both B and T cell responses and functions

To determine the effect of TLR9 on B and T cell function, we stimulated splenocytes from KO and Ctr mice with either anti-IgM together with anti-CD40 (B cell stimulation), or anti-CD3 together with anti-CD28 (T cell stimulation). We found that B cells deficient in *Tlr9* showed impaired proliferative responses to B cell receptor and costimulatory molecule stimulation (Fig. 3A). Interestingly, the proliferative T cell response to TCR and costimulatory molecule stimulation was significantly enhanced in the KO group compared to their control counterparts (Fig. 3B). Next, we assessed the proliferative responses of splenocytes to different innate immune stimuli, including Pam3Csk4, lipopolysaccharide (LPS), and CpG (TLR2, TLR4, and TLR9 agonists, respectively). B cells express high levels of TLRs, and we found that the splenocytes from the KO mice showed impaired responses to the innate immune stimuli, LPS and CpG, but normal responses to the TLR2 agonist Pam3Csk4 (Fig. 3C–E). Assessing concomitant changes in cytokine secretion, we observed significantly lower IL-10 in the culture supernatants following anti-IgM and anti-CD40 stimulation of B cells from KO mice compared with Ctr mice (Fig. 3F). However, in line with higher proliferative responses to T cell stimulation, the culture supernatants from KO mice had significantly higher levels of IFN-γ and IL-17a (Fig. 3G, H) in response to T cell stimulation, suggesting that the enhanced function of T cells may contribute to a pro-inflammatory environment.

To verify the specific function of B cells, we purified splenic B cells from KO and Ctr mice and stimulated the B cells using the same agonists. Supporting the results using total splenocytes, we found impaired B cell responses to both adaptive (anti-IgM and anti-CD40) and innate immune (CpG and Pam3Csk4) stimuli using purified B cells from KO mice (Fig. 3I–K). However, the B cell response to LPS was the opposite to the response using total splenocytes (Fig. 3C), where we observed stronger responses by B cells from KO mice (Fig. 3L).

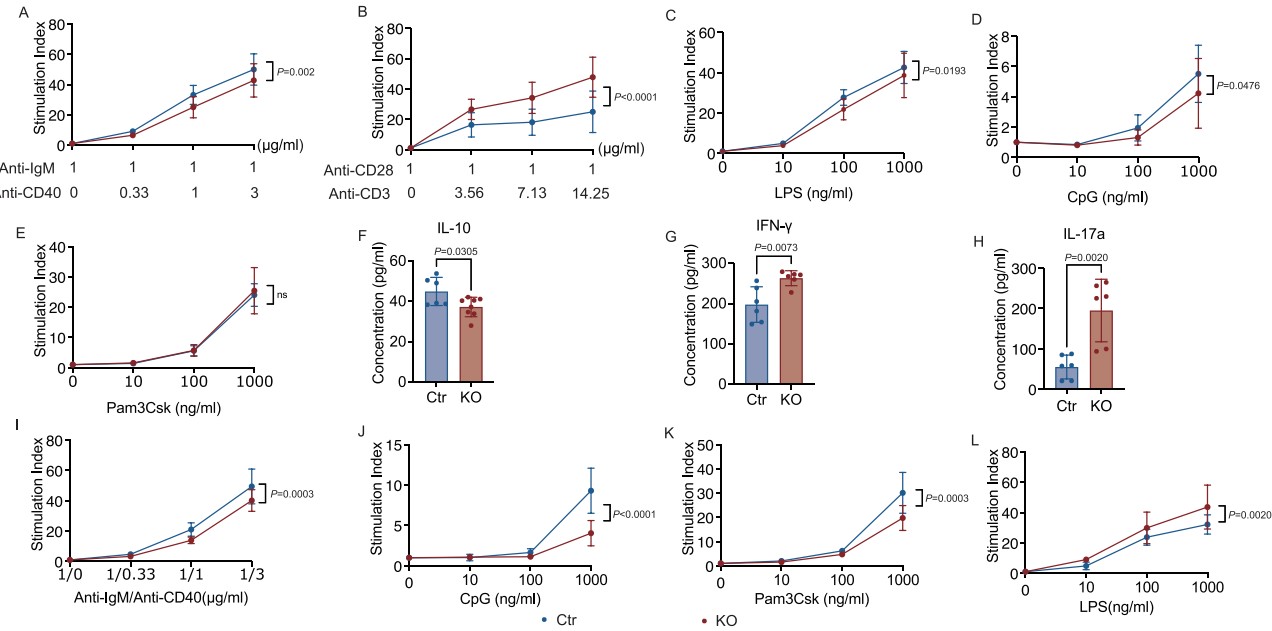

**Fig. 3 | *Tlr9* deficiency in B cells alters both B and T cell responses and function.** Immune cell proliferative response was assessed by ³H-thymidine incorporation assay. Immune cells were cultured for 3 days in the presence of different stimulators prior to adding ³H-thymidine. The cells were further cultured for 16-18 h before harvest. The proliferative results are presented as stimulation index, using counts per minute (CPM) in the presence of the stimulators divided by the background (CPM in the absence of the stimulators). Splenic immune cells from the KO and Ctr male mice were stimulated in the presence of anti-CD40 (different concentrations) together with anti-IgM (1μg/ml) (KO: *n* = 9 samples from 3 mice, Ctr: *n* = 12 samples from 4 mice) **A**, anti-CD3 (different dilutions of mAb) and anti-CD28 (1μg/ml) (KO: *n* = 12 samples from 4 mice, Ctr: *n* = 9 samples from 3 mice) **B**, LPS (KO: *n* = 9 samples from 3 mice, Ctr group: *n* = 12 samples from 4 mice) **C**, CpG (KO: *n* = 9 samples from 3 mice, Ctr: *n* = 12 samples from 4 mice) **D** and Pam3Csk4 (KO: *n* = 9 samples from 3 mice, Ctr: *n* = 12 samples from 4 mice) **E**. **F** Secreted IL-10 in the supernatants collected from the cultures with anti-CD40 and anti-IgM at 72 h (KO: *n* = 8 samples from 4 mice, Ctr: *n* = 6 samples from 3 mice). **G, H** Secreted IFN-γ and IL-17a in the supernatants collected from the cultures with anti-CD3 and anti-CD28 at 72 h (*n* = 6 samples from 3 mice per group). Purified splenic B cells from the KO and Ctr male mice were stimulated with different concentrations of anti-CD40 and anti-IgM **I**, CpG **J**, Pam3Csk4 **K**, and LPS **L**. (*n* = 9 samples from 3 mice per group). Total splenocytes were used for T cell stimulation (anti-CD3 and anti-CD28). All the experiments are repeated twice with similar results. Data in **A–E** and **I–L** were analyzed using two-way ANOVA. Data in **F–H** were analyzed using two-tailed Student's *t*-test. The data are presented as mean ± SD.

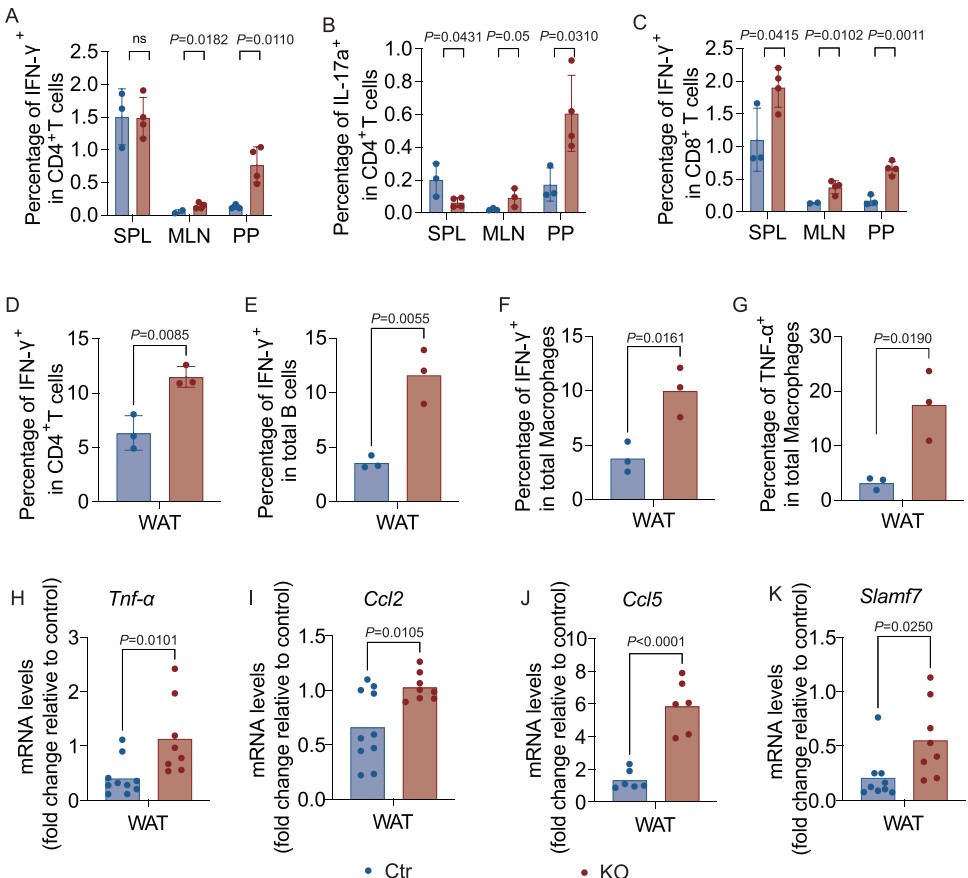

**Fig. 4 | *Tlr9* deficiency in B cells results in inflammatory T cells in lymphoid tissues and white adipose tissues (WAT).** Immune cells from different lymphoid tissues and WAT were prepared from male KO and Ctr mice ex vivo (at 18–22-week-old). **A–C** The cytokine profiles of the immune cells from different lymphoid tissues of the KO and Ctr mice, ex-vivo, were assessed by intra-cellular cytokine staining and flow cytometric analysis. TCR-β⁺ cells were gated as T cells. **A** Proportion of IFN-γ expressing CD4⁺ T cells (KO: *n* = 4 mice, Ctr: *n* = 3 mice). **B** Proportion of IL-17a-expressing CD4⁺ T cells (KO: *n* = 4 mice, Ctr: *n* = 3 mice). **C** Proportion of IFN-γ-expressing CD8⁺ T cells (KO: *n* = 4 mice, Ctr: *n* = 3 mice). **D–G** Immune cells from WAT of the male KO and Ctr mice were isolated and studied ex-vivo. The cytokine profiles of the WAT immune cells were assessed by intra-cellular cytokine staining and flow cytometric analysis (*n* = 3 samples from 6 mice per group). **D** Frequency of IFN-γ-expressing CD4⁺ T cells. **E** Frequency of IFN-γ-expressing B cells. **F** Frequency of IFN-γ-expressing macrophages. **G** Frequency of TNF-α-expressing macrophages. The cell numbers of **A–G** are shown in Supplementary Fig. 6. **H–K** Inflammatory cytokine and chemokines in WAT were assessed by qPCR. Total RNA was extracted from WAT of 18–22-week-old male KO and Ctr mice after 12-16 weeks of HFD and qPCR was performed with specific primers. The reference gene, *Gapdh* was utilized as an internal reference for normalization. The relative gene expression levels were calculated using $2^{-\Delta\Delta Ct}$ method by normalization with the reference gene *Gapdh*. **H–K** *Tnf-α*, *Ccl2*, *Ccl5* and *Slamf7* expression levels in WAT (KO: *n* = 6–8 mice, Ctr: *n* = 6–10 mice). All the experiments are repeated twice with similar results. Data were first run through normal distribution test before analyzing with two-tailed Student's *t*-test. The data are presented as mean ± SD.

Collectively, these studies outline a critical role of TLR9 in B cells, which affect B cell function (reduced proliferation, and altered cytokine production) accompanied by enhanced T cell function toward an inflammatory milieu.

## Tlr9 deficiency in B cells results in increased inflammatory T cells in lymphoid tissues and white adipose tissues (WAT)

As our data in the mice on normal diet suggested that B cells, in the absence of *Tlr9*, promoted inflammatory T cells after T cell stimulation, we next investigated the cytokine profiles of the immune cells from different lymphoid tissues of the HFD-fed mice. We found that the IFN-γ-producing CD4⁺ T cells, IL-17a-producing CD4⁺ T cells and IFN-γ-producing CD8⁺ T cells were higher in MLNs and PPs from KO mice, compared with the Ctr mice (Fig. 4A–C and Fig.S6A–C). The pro-inflammatory cytokine-producing T cells were more striking in PPs, suggesting an inflammatory environment in the intestine.

We then assessed the immune cells from WAT tissue. Supporting our data presented earlier, all the immune cells examined in the WAT from KO mice had a significantly higher proportion and number of IFN-γ-producing CD4⁺ T-cells, B cells and macrophages, compared to the

controls (Fig. 4D–F, Fig. S6D–F). A higher percentage of the macrophages from WAT of KO mice also produced TNF-α (Fig. 4G and Fig. S6G). To further verify that the increased obesity found in KO mice after HFD diet was associated with inflammation in adipocytes, we examined inflammatory markers in the adipose tissue, liver and muscle by qPCR. We tested various inflammatory markers, and among them found significant increases in *Tnf-α*, inflammatory chemokines *Ccl2*, *Ccl5, and Slamf7* in the adipose tissue from KO mice compared to the adipose tissue from the Ctr mice (Fig. 4H–K). However, no significant differences in the inflammatory markers were found in liver and muscle from the two groups.

## Tlr9 deficiency in B cells alters the mucosal immunoglobulin levels and gut microbiota communities

One of the major functions of B cells is to produce immunoglobulins (Ig), which specifically recognize and bind to antigens from pathogens, microbial molecules and sometimes host self-antigens. Immunoglobulins, especially IgA, play an important role in maintaining a healthy balance between the host and the gut commensal bacteria[48,49]. To investigate whether B cell-specific deletion of *Tlr9* altered intestinal

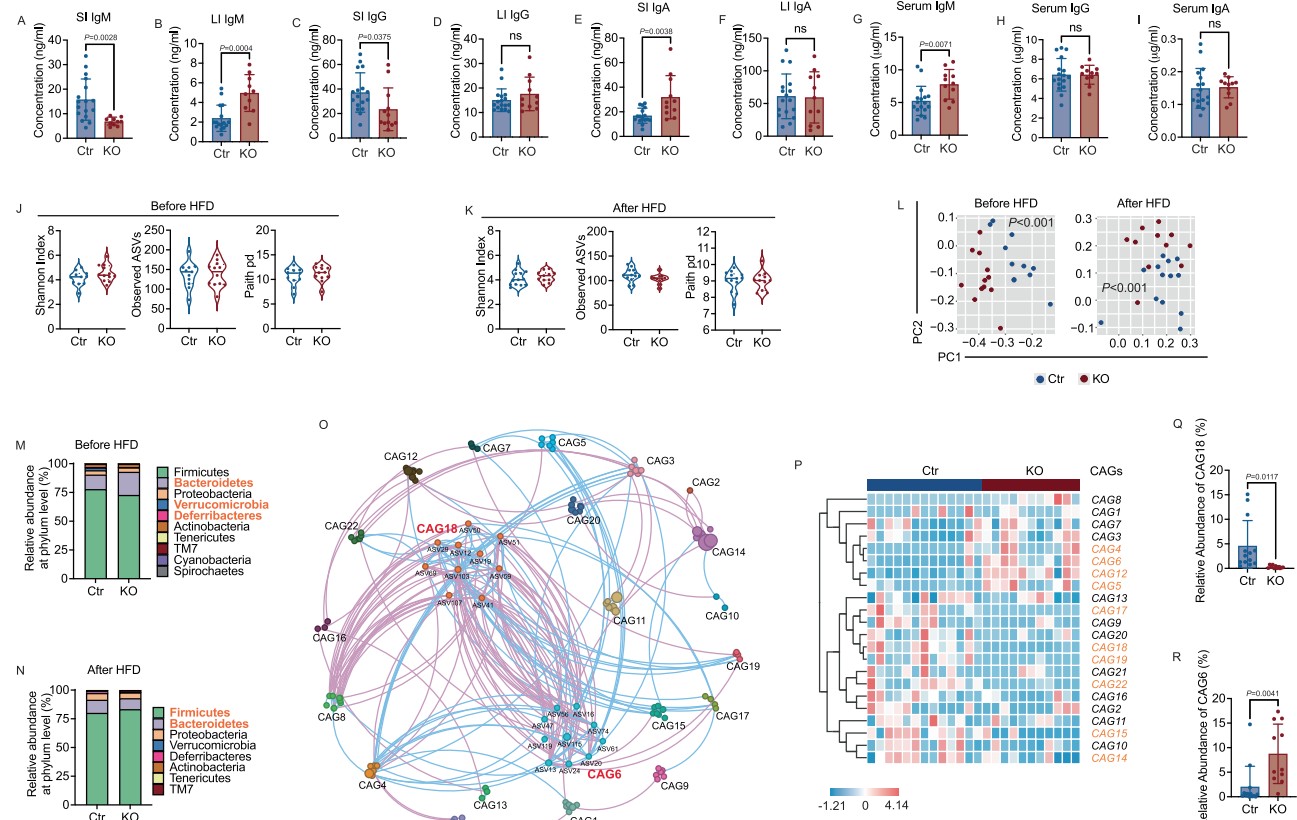

**Fig. 5 | *Tlr9* deficiency in B cells changes the levels of mucosal immunoglobulins and gut microbiota composition.** Immunoglobulins (IgM, IgG and IgA) from intestinal flush and serum samples from KO and Ctr mice, all males, after HFD were assessed by ELISA. Fecal pellets were collected from male KO and Ctr male mice before and after 12-week feeding with HFD. Bacterial DNA was extracted and the microbial composition was assessed by 16 S rRNA gene deep sequencing. **A–F** Concentrations of IgM, IgG, IgA in small intestinal gut flush (SI) and large intestinal gut flush (LI) from 18-22-week-old KO and Ctr male mice were determined by ELISA with a standard curve for each Ig (KO: *n* = 10–12 mice, Ctr: *n* = 17 mice). **G–I** Concentrations of IgM, IgG, IgA in serum from the same KO and Ctr mice (KO: *n* = 11 mice, Ctr: *n* = 17 mice). **J, K** α diversity (Shannon Index, Observed ASVs and Paith pd) of gut microbiota in the mice before and after HFD. **L** The overall structure of the gut microbiota in both Ctr and KO group mice, before and after HFD, was assessed. Principal coordinate analysis (PCoA) was carried out using Bray-Curtis distance metrics at the Amplicon Sequence Variants (ASVs) level. **M, N** The relative

abundance of gut microbiota at the phylum level in the KO group and Ctr group before and after HFD. The phyla marked in bold orange indicate significant differences between the two groups. **J–N** Before HFD: KO: *n* = 12 mice, Ctr: *n* = 11 mice. After HFD: KO: *n* = 11 mice, Ctr: *n* = 13 mice. **O** Microbial interaction network displays the interaction between different Co-abundant groups (CAGs) after HFD. The size of node represents the mean abundance of ASV, being larger size, higher abundance. The lines between the nodes indicate correlation (pink = negative correlation, blue = positive correlation), width of the lines corresponding to magnitude of the correlation. **P** The heatmap of CAGs in Ctr group and KO group after HFD. The CAG number highlighted in bold orange indicate significant differences between the two groups by PERMANOVA. **Q, R** The relative abundance of CAG6 and CAG18 between Ctr and KO group (KO: *n* = 11 mice, Ctr: *n* = 13 mice). The data in **A–I** are shown as the mean ± SD, and Student's t-test (two-tailed) was used to analyze difference between Ctr and KO groups. The experiment was performed 3 times and the pooled results are presented in the figure.

antibodies, we measured immunoglobulins in small and large intestinal gut flush. We found that the levels of IgM in small intestine were significantly decreased in the KO group compared to the Ctr group (Fig. 5A), whereas IgM in the large intestine was significantly increased (Fig. 5B). Similarly, IgG in small intestine of the KO group was also significantly reduced compared to the Ctr group (Fig. 5C). Unlike the large intestinal IgM, IgG levels in the large intestine were comparable between the two groups of mice (Fig. 5D). IgA, the major antibody in mucosal sites, was significantly increased in the small intestine but there were no differences in large intestine (Fig. 5E, F). Our flow cytometry results from PP indicated that both GC B cells and T$_{FH}$ cells were increased in KO mice compared to controls, which likely contributed to the increase in IgA. Next, we assessed Ig levels in the serum samples of the same mice. IgM in serum was also higher in the KO group compared to the Ctr group (Fig. 5G), whereas no differences were found in serum IgG and IgA (Fig. 5H, I). Thus, the IgM response was systemic whereas IgG and IgA responses were compartmentalized in the intestine.

As mucosal antibodies play an important role in the homeostasis of gut microbiota, we asked whether *Tlr9* deficiency in B cells would alter the composition of gut microbiota. We isolated bacterial DNA and performed 16 S rRNA sequencing of fecal samples from KO and Ctr mice before and after high fat diet. Although we did not find significant differences in alpha diversity and richness between the two groups of mice (Fig. 5J, K), the structure of gut microbiota in the KO mice was significantly different from the control mice as seen in the principal coordinate analysis (PCoA) plot of Bray-Curtis distance [*P* < 0.001 with permutational multivariate analysis of variance (PERMANOVA) test] in both normal and high-fat diet conditions (Fig. 5L). The ratio of the relative abundance of Firmicutes to Bacteroidetes (F/B) has been reported to be significantly associated with obesity[50,51]. The F/B ratio in the gut microbiota of the KO mice was lower than that of Ctr mice under normal diet conditions as there were more Bacteroidetes in KO mice (Fig. 5M). However, the F/B ratio was significantly increased after high-fat diet (Fig. 5N). This may suggest that the gut microbiota of the KO mice are more susceptible to environmental stress, such as a high-

fat diet. To identify the key members of the gut microbiota that may contribute to the metabolic phenotype in the KO mice, we used co-abundance analysis to identify potentially functional bacterial groups. The microbiota structure of the mice on normal diet (ND) and high-fat diet (HFD) was analyzed separately to prevent any possible influence of Simpson's paradox. The microbial co-abundance network was built to visualize the possible interactions among the 125 Amplicon Sequence Variants (ASVs) that shared at least 20% of the samples. The ASVs were further clustered into 32 co-abundant groups (CAGs) by the SparCC method and PERMANOVA analysis[52]. From the spatial topological map, we observed a highly significant negative correlation (purple lines) between CAG18 and CAG6 within the ecological community of the mouse gut microbiota with high-fat diet (Fig. 5O). However, this negative correlation was not observed under normal diet conditions (Fig. S7A). The CAGs with significant differences between the two groups on HFD were highlighted in orange (10/22, 45.45%; Fig. 5P). Whereas there were fewer number of CAGs that were significantly different between the two groups in the normal diet (9/28, 32%; Fig. S7B). Further, analysis revealed that there were 10 ASVs in each CAG6 and CAG18. Among the 10 ASVs in CAG18, 8 belonged to Lachnospiraceae. Interestingly, we found that CAG18 was significantly decreased in the KO group (Fig. 5Q), whereas CAG6, exhibiting a significant negative correlation with CAG18, was significantly enriched in the KO group (Fig. 5R). The decrease in strains comprising CAG18 or the increase in strains comprising CAG6 were likely to be the bacterial communities contributing to the metabolic phenotype seen in the KO mice. In the CAGs analysis, the complete matrix for HFD is available in Supplement Data 1, and the complete matrix for ND is available in Supplement data 2. Taken together, our data provide important insights into *Tlr9* deficiency in B cells in the context of altered gut microbiota in obesity. Thus, we have shown that lack of TLR9 in B cells not only affected systemic B and T cell phenotypes and function but there were also altered gut mucosal antibody profiles and gut microbiota composition, which are closely associated with obesity.

### Altered gut microbiota from Tlr9^fl/fl^/Cd19Cre^+/-^ mice can transfer immunological and metabolic phenotypes to germ-free (GF) mice

To verify whether the altered gut microbiota contributed to the increased body weight, impaired metabolic function, and heightened inflammation seen in the KO mice, we performed fecal microbiota transfer (FMT) experiments (Fig. 6A). The respective pooled stool samples from KO and Ctr mice (18–22-week-old, 12–16 weeks post HFD) were used to colonize GF wild type B6 mice (https://www.jax.org/strain/000664, rederived to GF by Dr. Flavell, Yale) by oral gavage (at 4 weeks of age). Colonized ex-GF mice were maintained on standard diet and their IPGTT was assessed on day 10 after FMT. We found that the gut microbiota from KO donors induced impaired glucose tolerance in the recipient mice (Fig. 6B). The AUC of the IPGTT was also greater in KO group (Fig. 6C). Moreover, the gut microbiota from KO donors modulated B and T cells in the recipient mice (Fig. 6D–K and Fig. S8A–H). The proportion and the number of Breg (CD5+CD1d+ B cells) were decreased in MLNs (Fig. 6D and Fig. S8A). Interestingly, the GC B cells (IgD^low^CD95+GL7+ B cells) were also markedly decreased in PPs (Fig. 6E and Fig. S8B). We found an increase in total T cells (Fig. 6F and Fig. S8C). The proportion and number of both T_FH_ (CXCR5+PD-1+CD4+) cells and T_PH_ (CXCR5-PD-1+CD4+) cells were significantly higher in mucosa-associated MLNs and PPs but not in SPL (Fig. 6G, H and Fig. S8D, E). IL-10-producing B cells were significantly decreased in the PPs of the GF mice that received fecal microbiota from KO donor mice compared to the Ctr donor mice (Fig. 6I and Fig. S8F). In line with the phenotype of KO donor mice (Fig. 4A, B), the IFN-γ and IL-17a-producing CD4+ T cells were significantly increased in the PPs of ex-GF recipient mice (Fig. 6J, K and Fig. S8G, H). To assess whether changes observed from the short-term FMT transfer approach would remain beyond 14 days, we repeated the experiment with a longer observation period of 5 weeks after FMT (Fig. 6L). Supporting the results from the short-term FMT experiments, the body weights of ex-GF mice that received FMT from KO donor mice were significantly higher than the control counterparts (Fig. 6M, N). Also, in line with the results from the short-term FMT experiment, the gut microbiota from KO donors induced impaired glucose tolerance 4 weeks after FMT in the ex-GF mice (Fig. 6O), and significant higher AUC values (Fig. 6P). Unlike the results from the short-term FMT experiment, the phenotypes of B and T cells in the lymphoid tissues examined were comparable between the 2 groups (data not shown). To verify the direct effect of different gut microbiota from KO group and Ctr group on the immune cells, we performed in vitro co-culture experiments. We cocultured spleen cells from wild type B6 mice with stool bacteria from the KO and Ctr mice overnight and assessed the phenotypes of T cells. Similar to the T cell phenotypes found in the KO mice ex vivo (Fig. 2G–M), the splenic T cells from wild type B6 mice showed a significant increase, in vitro, in the proportion of T cells (Fig. 6Q), T_FH_ cells (Fig. 6R) and T_PH_ cells (Fig. 6S), when stimulated with the gut microbiota from KO mice, compared with the gut microbiota from Ctr mice. Moreover, the gut microbiota from KO mice promoted immune cells producing more IFN-γ (Fig. 6T) but reduced IL-10 production (Fig. 6U).

Our data suggest that not only can the gut microbiota from the *Tlr9^fl/fl^/Cd19Cre^+/-^* B6 mice transfer glucose metabolic dysfunction to the GF mice, in a period as short as 10 days or as long as 4 weeks after FMT, but this transfer also had both indirect and direct effect on the immune cells.

### Tlr9 Deficiency in B cells leads to metabolic disturbances in immune cells and the gut microbiome

As we had shown that *Tlr9* deficiency in B cells altered host metabolism and immune cells, to further identify the role of immune cells in host metabolism, we adoptively transferred splenic lymphocytes from KO and Ctr donor mice to 4-week-old SPF *Rag1^-/-^* B6 mice (10^7^/mouse), followed by administering HFD when the recipient mice were 6 weeks old (Fig. 7A). As expected, body weight was much higher in the recipient SPF *Rag1^-/-^* B6 mice that received immune cells from KO donors (Fig. 7B, C). Moreover, those recipient SPF *Rag1^-/-^* B6 mice had higher fasting glucose levels (Fig. 7D), impaired glucose tolerance (Fig. 7E), higher AUC values in the IPGTT (Fig. 7F). Those mice were also insulin resistant (Fig. 7G) with higher AUC values in the ITT (Fig. 7H) compared to the mice infused with the immune cells from the Ctr donors.

Based on the impact of *Tlr9* deficiency in B cells on metabolic function and the gut microbiota impact on both immune system and metabolism from different experiments, we further investigated whether specific gut bacteria were responsible for the metabolic changes. We performed a set of experiments with criss-cross experimental design, using different combination of gut bacteria and immune cells to germ-free *Rag1^-/-^* B6 recipient mice. This involved intravenously infusing splenic immune cells from KO donor mice together with orally-gavaged gut bacteria from KO or Ctr donor mice, comparing these with infusions of immune cells from Ctr donor mice and together with orally-gavaged gut bacteria from KO or Ctr donor mice. The mice were observed for 4 weeks (Fig. 7I). IPGTT showed that the germ-free *Rag1^-/-^* B6 recipient mice, which were transferred with spleen cells from KO donors had more impaired glucose tolerance compared with the recipient mice transferred with spleen cells from Ctr donor mice (Fig. 7J). Furthermore, the recipient mice that were transferred with spleen cells from KO donors, together with the gut bacteria from the KO donors had more impaired glucose tolerance than the recipient mice that received spleen cells from KO donors but bacteria from Ctr donors (Fig. 7J). The AUC of IPGTT also indicated that the highest AUC was seen in the germ-free *Rag1^-/-^* B6 recipients of both immune cells and gut bacteria from the KO donors (Fig. 7K). Thus, our results suggest that B cell dysfunction and altered gut microbiota are both required to cause the metabolic

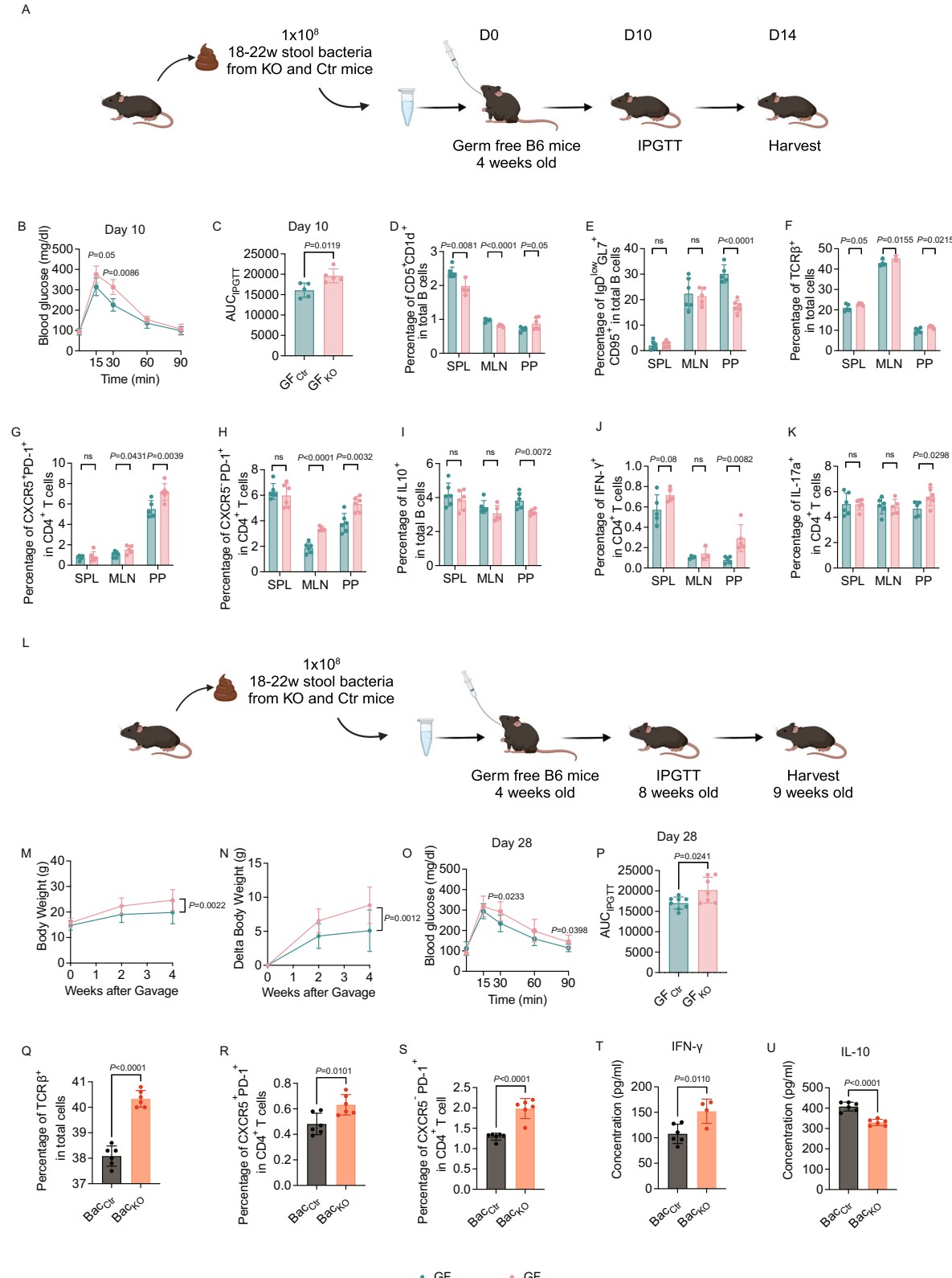

changes, and *Tlr9* deficiency in B cells appears to have a stronger effect on the metabolic dysfunction in the host, and the altered gut microbiota enhanced the metabolic changes.

## Tlr9 regulates B cell function through Irf4 and Il-10

To determine the molecular mechanism by which the *Tlr9*-deficient B cells altered the immune system, we performed total RNAseq with

purified splenic B cells from *Tlr9fl/fl/Cd19Cre+/-* and *Tlr9fl/fl/Cd19Cre-/-* B6 mice. The principal component analysis showed highly significant separation of the two groups of B cells, indicating that the transcriptomes of the two groups of the B cells were very different (Fig. 8A). The volcano plot and the heatmap showed that more genes had decreased rather than increased expression in B cells from KO mice compared with Ctr mice (Fig. 8B, C). Interferon regulatory factor

**Fig. 6 | Altered gut microbiota from *Tlr9^fl/fl^/Cd19Cre^+/-^* B6 mice can transfer the immunological and metabolic phenotype to germ-free (GF) mice in vivo and influence immune cells in vitro.** Microbiota from pooled fecal samples collected from the KO and the Ctr male mice after 12–16 wks of HFD were used to colonize wild type GF B6 mice (4 weeks of age) by oral gavage. **A** The experimental procedure of fecal microbiota transfer (FMT) to germ free (GF) B6 mice (short-term, 2 weeks). GF wild type B6 mice (4 weeks old) were gavaged with $1 \times 10^8$ cfu gut microbiota from the pooled stool samples of *Tlr9^fl/fl^/Cd19Cre^+/-^* (KO) and *Tlr9^fl/fl^/Cd19Cre^-/-^* B6 (Ctr) male mice. The recipient GF B6 mice were maintained on normal diet. IPGTT was assessed 10 days post FMT. The experiment was terminated 14 days after FMT. Graph was created with BioRender.com. **B** IPGTT results of the ex-GF B6 mice, 10 days post gavage (*n* = 5 mice per group). **C** Area under the curve (AUC) of the IPGTT results from **A** (*n* = 5 mice per group). **D–K** Immune cell phenotype of ex-GF B6 mice assessed by flow cytometric analysis, 14 days after gut microbiota colonization. **D** Proportion of CD5^+^CD1d^+^ B cells (gated on B220^+^) (*n* = 6 mice per group). **E** Proportion of IgD^low^GL7^+^CD95^+^ B cells (*n* = 6 mice per group). **F** Proportion of total TCRαβ^+^ T cells (*n* = 5 mice per group). **G** Proportion of CXCR5^-^PD-1^+^CD4^+^ T cells (*n* = 6 mice per group). **H** Proportion of CXCR5^-^PD-1^-^ CD4^+^ T cells (*n* = 6 mice per group). **I** Proportion of IL-10-expressing B cells (*n* = 6 mice per group). **J** Proportion of IFN-γ-expressing CD4^+^ T cells (*n* = 5 mice per group). **K** Proportion of IL-17a-expressing CD4 T^+^ cells (*n* = 6 mice per group). **L** The experimental procedure of fecal microbiota transfer (FMT) to GF B6 mice (long-

term, 4 weeks). GF wild type B6 mice (4 weeks old) were gavaged with $1 \times 10^8$ cfu gut microbiota from the pooled stool samples of *Tlr9^fl/fl^/Cd19Cre^+/-^* (KO) and *Tlr9^fl/fl^/Cd19Cre^-/-^* B6 (Ctr) male mice (12–16 weeks post HFD). The recipient GF B6 mice were maintained on normal diet. IPGTT was assessed 4 weeks after FMT. The experiment was terminated 5 weeks after FMT. Graph was created with BioRender.com. **M** Body weight change of the ex-GF B6 mice colonized with gut microbiota from KO (GF_KO) and control mice (GF_Ctr) (GF_KO: n = 7 mice, GF_Ctr: *n* = 9 mice). **N** Net body weight gain of the ex-GF B6 mice colonized with gut microbiota from *Tlr9^fl/fl^/Cd19Cre^+/-^* B6 mice or *Tlr9^fl/fl^/Cd19Cre^-/-^* B6 mice at 4 weeks post gavage (GF_KO: n = 7 mice, GF_Ctr: *n* = 9 mice). **O** IPGTT results of the ex-GF B6 mice 4 weeks after colonization with gut microbiota from KO and Ctr mice (GF_KO: *n* = 7 mice, GF_Ctr: *n* = 8 mice). **P** Area under the curve (AUC) of the IPGTT results in **M** (GF_KO: *n* = 7 mice, GF_Ctr: *n* = 8 mice). **Q–U** Gut microbiota from B cell *Tlr9*-deficient mice affect immune cell phenotype and cytokine profile in vitro. **Q** Frequency of total T cells after culture (*n* = 6 wells per group). **R** Frequency of CXCR5^+^PD-1^+^CD4^+^ T cells after culture (*n* = 6 wells per group). **S** Frequency of CXCR5^-^PD-1^+^CD4^+^ T cells after culture (*n* = 6 wells per group). **T** Secreted IFN-γ levels in the culture supernatants (Bac_KO: n = 4 wells, Bac_Ctr: *n* = 6 wells). **U** Secreted IL-10 levels in the culture supernatants (*n* = 6 samples per group). All the data were repeated twice with similar results. K&L were analyzed using two-way ANOVA. Other data were analyzed using two-tailed Student's *t*-test. The data are presented as mean ± SD. The cell numbers of **D–K** are shown in Supplementary Fig. 8.

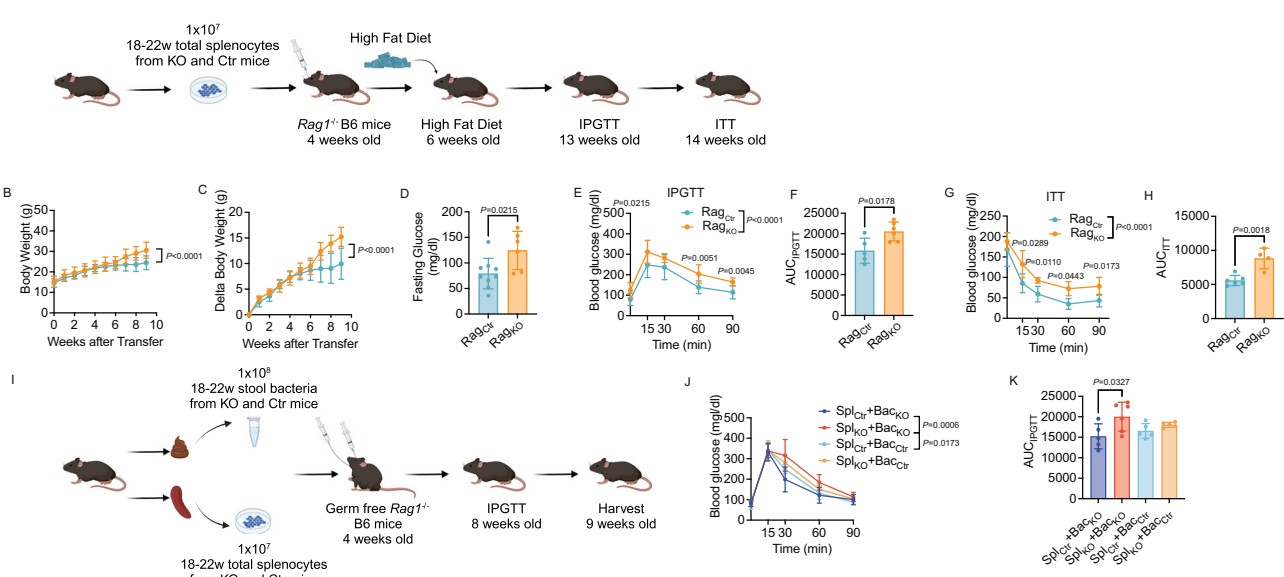

**Fig. 7 | *Tlr9* deficiency in B cells leads to a metabolic disorder in immunodeficient Rag^-/-^ B6 mice mediated by the immune system and gut microbiota.** SPF *Rag1^-/-^* and GF *Rag1^-/-^* B6 mice were recipients for transfer of immune cells or/and gut microbiota from KO and Ctr donor mice. **A** The experimental procedure of immune cell adoptive transfer to SPF *Rag1^-/-^* B6 mice. SPF *Rag1^-/-^* B6 mice (4 weeks of age) were adoptively transferred (i.v.) $1 \times 10^7$ total splenocytes from *Tlr9^fl/fl^/Cd19Cre^+/-^* (KO) and *Tlr9^fl/fl^/Cd19Cre^-/-^* B6 (Ctr) male donor mice (12–16 weeks post HFD). Two weeks after the immune cell transfer, the recipient *Rag1^-/-^* B6 mice (6 wks of age) were started with high fat diet. IPGTT and ITT were assessed when the mice were at 13 and 14 weeks of age, respectively. Graph was created with BioRender.com. **B, C** Body weight change and net body weight gain of the SPF *Rag1^-/-^* B6 mice transferred with the immune cells from KO and Ctr donors (Rag_KO: *n* = 7 mice, Rag_Ctr: *n* = 8 mice). **D** Fasting glucose of the SPF *Rag1^-/-^* B6 recipient mice transferred with the immune cells from KO and control mice (Rag_KO: *n* = 9 mice, Rag_Ctr: *n* = 6 mice). **E** IPGTT result of the SPF *Rag1^-/-^* B6 recipient mice transferred with immune cells from KO and Ctr donor mice (Rag_KO: *n* = 9 mice, Rag_Ctr: *n* = 6 mice). **F** Area under the curve (AUC) of IPGTT results from D. **G** ITT result of the SPF *Rag1^-/-^*

recipient B6 mice transferred with immune cells from KO and control donors (Rag_KO: *n* = 4 mice, Rag_Ctr: *n* = 6 mice). **H** Area under the curve (AUC) of the ITT of the results from **F**. **I** The experimental procedure of immune cell adoptive transfer and fecal microbiota transfer (FMT) to germ free *Rag1^-/-^* B6 mice. Germ free *Rag1^-/-^* B6 mice (4 weeks of age) were adoptively transferred (i.v.) $1 \times 10^7$ total splenocytes and gavaged with $1 \times 10^8$ gut microbiota from the pooled stool samples from *Tlr9^fl/fl^/Cd19Cre^+/-^* (KO) or *Tlr9^fl/fl^/Cd19Cre^-/-^* B6 (Ctr) male donor mice (12–16 weeks post HFD). The recipient mice were maintained on normal diet. IPGTT was assessed 4 weeks after the FMT and immune cell transfer. The experiment was terminated 5 weeks after the FMT and immune cell transfer. Graph was created with BioRender.com. **J** IPGTT result of GF *Rag1^-/-^* B6 mice transferred with the immune cells from KO and Ctr donors and colonized with the bacteria from the same KO and Ctr donor mice (*n* = 5 mice per group). **K** Area under the curve (AUC) of the IPGTT of the results from **J**. All the experiments were repeated twice with similar results. **B, C, E, G, J** were analyzed using two-way ANOVA. **K** was analyzed using one-way ANOVA. **D, F, H** were analyzed using two-tailed Student's *t*-test. The data are presented as mean ± SD.

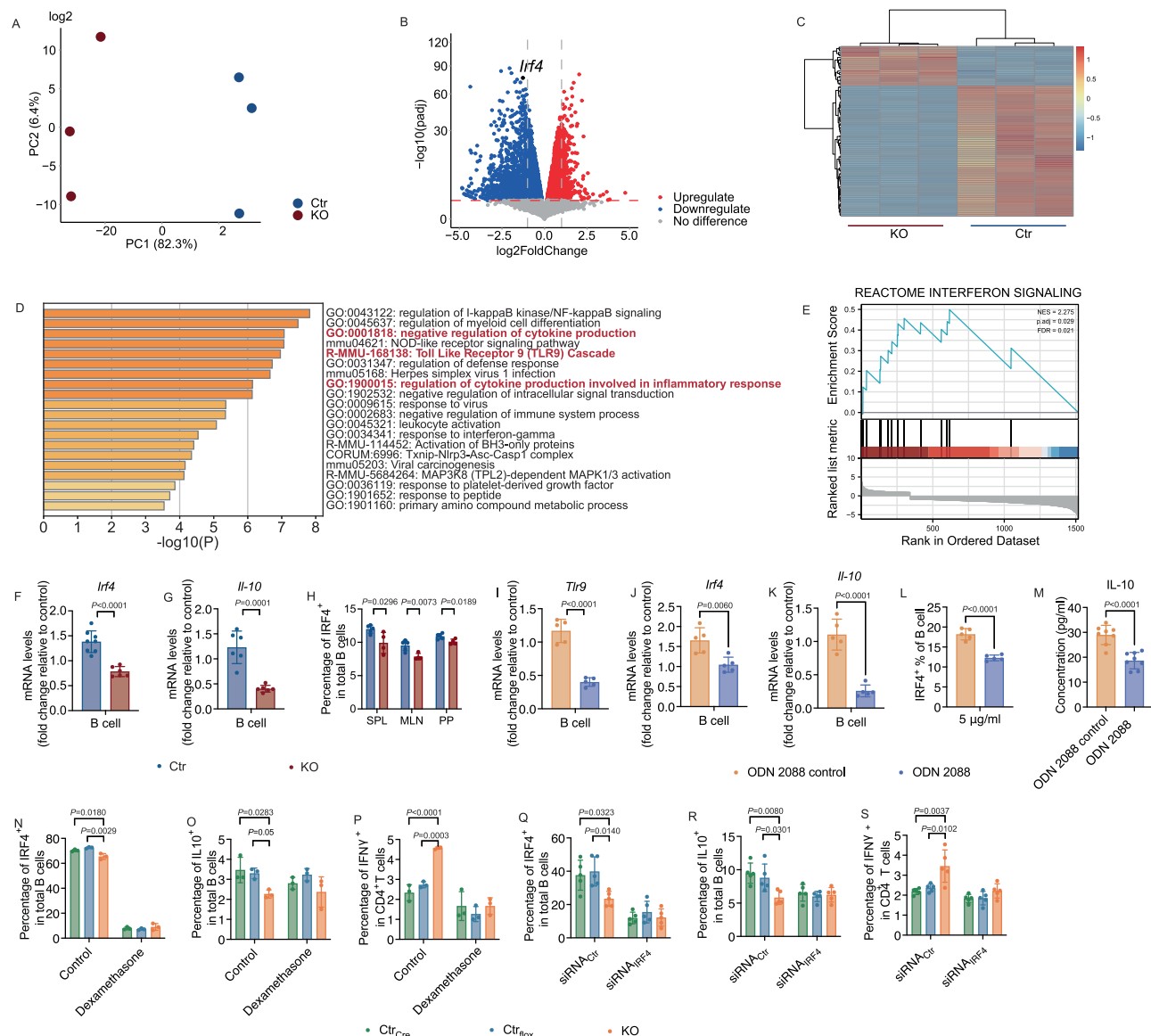

**Fig. 8 | *Tlr9* regulates B cell function through *Irf4* and *Il-10*.** Total RNA sequencing analysis was performed using purified B cells from 18-22-week-old male KO and Ctr mice (*n* = 3 mice per group). **A** Principal component analysis plot is shown comparing the sample diversity between the two groups. Blue and red represent the samples from Ctr and KO mice, respectively. **B** Volcano plot, differential gene expression (DGE) analysis was performed to compare combined gene expression in B cells from the two groups. Red represents up-regulated and blue represent down-regulated gene expressions (adjust *p* < 0.05). **C** Heatmap of top 100 differential genes, up-regulated and down-regulated genes were selected and clustered according to Gene Ontology analysis. The B cells from KO mice have more differentially down-regulated genes whereas the B cells from control mice have more differentially up-regulated genes. **D** Analysis of selected GO parent enrichment functions. **E** Analysis of GSEA (Gene Set Enrichment Analysis) pathway. NES = 2.275, p. adj = 0.029, FDR = 0.021. **F** *Irf4* expression level in purified B cells by qPCR (KO: n = 6 mice, Ctr: *n* = 8 mice). **G** *Il-10* expression level in purified B cells tested by qPCR (*n* = 6 mice per group). **H** Proportion of IRF4-expressing B cells from KO and Ctr mice by flow cytometric analysis (KO: *n* = 4 mice, Ctr: *n* = 5 mice). **I**–**M** Inhibition of TLR9 in purified B cells from control B6 mice recapitulates the phenotypic features of KO mice. Purified splenocytes from control B6 mice were cultured overnight in the presence of the TLR9 antagonist, ODN2088 and ODN2088 control. The cells were collected and washed post culture for qPCR and flow cytometric analysis. **I**–**K** *Tlr9*, *Irf4* and *Il-10* expression levels in the cultured B cells analyzed using qPCR

(*n* = 5 wells per group). **L** The frequency of IRF4-expressing B cells by flow cytometric analysis after culture (*n* = 5 wells per group). **M** Secreted IL-10 levels in the supernatants of the cultured cells by ELISA (*n* = 8 wells per group). **N**–**P** Inhibition of IRF4 abolished the phenotype of reduction of IL-10-producing Bregs and increase of IFNγ-CD4+ T cells. Splenocytes from KO and two Ctr groups (*Tlr9+/+/Cd19Cre+/-*, designated as Ctr_Cre, and *Tlr9fl/fl/Cd19Cre-/-*, designated as Ctr_flox) were cultured with solvent or dexamethasone (10 μM) overnight. Cultured cells were harvested, washed and analyzed using flow cytometry, after staining for different markers (*n* = 3 mice per group). **N** Proportion of IRF4 + B cells; **O** Proportion of IL-10-producing Breg cells. **P** Proportion of IFNγ-CD4+ T cells. All the data were from at least two independent experiments. **Q**–**S** IRF4-sepcific siRNA knock-down abolished the differences between KO and control mice in IL-10-producing Bregs and IFNγ-producing CD4+ T cells. Splenocytes from KO and two Ctr groups (Ctr_Cre and Ctr_flox) were cultured with siRNA_ctr or siRNA_IRF4 (1μM) for 48 h. Cultured cells were harvested, washed and analyzed using flow cytometry, after staining for different markers (*n* = 5 mice per group). **Q** Proportion of IRF4+ B cells; **R** Proportion of IL-10-producing Breg cells. **S** Proportion of IFNγ-CD4+ T cells. Experiments from F to M were carried out at least twice with similar results. Experiments N to S were performed in two independent experiments with similar results. Data are presented from one of the experiments. F-M were analyzed using two-tailed Student's *t*-test. N-S were analyzed using one-way ANOVA. The data are presented as mean ± SD.

4 (IRF4) was one of the highly decreased expressed genes. The transcription factor IRF4 plays a critical role in the regulation of the immune system[53,54]. It regulates inflammation by modulating the production of cytokines and chemokines, as well as regulating the differentiation and function of immune cells involved in inflammation[55–57]. IRF4 also regulates IL-10, which was significantly decreased in our KO mice compared to the controls (Fig. 2C, Fig S3C and Fig. 3F). The analysis of selected GO (Gene Ontology) parent heatmaps showed that there were several pathways regulating cytokine production involved in inflammatory responses (Fig. 8D). The GSEA (Gene Set Enrichment Analysis) also showed that the interferon signaling pathway was significantly involved in this process (Fig. 8E). Thus, these results suggested a link between *Tlr9* and *Irf4*. We further confirmed the expression of *Irf4* and *Il-10* transcripts by qRT-PCR using purified B cells (Fig. 8F, G). Moreover, we confirmed the low levels of IRF4+ B cells from different lymphoid tissues of KO mice, by flow cytometric analysis (Fig. 8H). To establish the link between *Tlr9* and *Irf4*, we treated purified splenic B cells from control mice with TLR9 antagonist (ODN2088) and a control for ODN2088 in in vitro culture overnight. The cells were collected for qPCR and flow cytometric analysis. The mRNA levels of *Tlr9*, *Irf4* and *Il-10* were all significantly decreased after treatment with TLR9 antagonist (Fig. 8I–K), as was the protein level of IRF4 in B cells (Fig. 8L). In addition, we assessed secreted IL-10 in the culture supernatants. In line with the qPCR results, the secreted IL-10 in the culture supernatant was also significantly lower after TLR9 antagonist treatment (Fig. 8M). To verify the role of IRF4 in IL-10 seen in KO mice, we treated the splenocytes from KO and Ctr mice with dexamethasone, a hormone that inhibits IRF4[58,59]. When splenocytes from KO and Ctr mice were treated with 10 μM dexamethasone separately in vitro, IRF4 was significantly suppressed in all the cells (Fig. 8N) confirming that dexamethasone indeed blocked IRF4. Moreover, dexamethasone abolished the differences found in KO mice, including decreased IL-10 producing Bregs and increased IFNγ-producing CD4+ T cells (Fig. 8O, P). To further verify the results using dexamethasone, we treated the splenocytes from KO and Ctr mice with IRF4-specific siRNA in vitro, and we found similar results as seen with dexamethasone treatment (Fig. 8Q–S). Thus, our data support a link between IRF4 and IL-10 in KO mice. Our in vitro experiments demonstrate that specific deletion of TLR9 in B cells affects the expression of IRF4 to some extent, thereby influencing the expression of IL-10. The experimental results agreed with the features that we observed in the animals, in vivo and ex-vivo. Our results also supported the notion that *Tlr9* regulated *Il-10* through *Irf4*, which could lead to inflammation in the host, together with the altered gut microbiota. Taken together, our data indicate that TLR9 in B cells plays an important role in metabolic dysregulation and obesity.

## Discussion

Recent studies suggest that TLR9 plays an important role in obesity and related inflammation and insulin resistance in humans and animals[31,37,38]. TLR9 modulates the ratio of M1/M2 macrophages, thereby impacting hepatic lipid metabolism in humans[60]. TLR9 regulates inflammatory responses in both mouse and human adipose tissue and influences lipid metabolism[61]. B cells express high levels of TLR9[62,63]; however, the role of TLR9 in B cells in the development of obesity has not previously been clear. To investigate the intrinsic role of TLR9 in B cells in obesity development, we generated a C57BL/6 mouse strain with B lymphocyte specifically deficient in *Tlr9* (*Tlr9fl/fl/Cd19Cre+/-* B6, KO). The Cre gene was knocked into the exon 2 of *cd19* gene and whereas homozygous *Cd19Cre* (*Cd19Cre+/+*) mice had impaired B cell development, for this study, we used *Cd19Cre* heterozygous (*Cd19Cre+/-*) mice in which B cell development was comparable to wild type (*Cd19Cre-/-*) mice[17,64]. We found that, in the absence of *Tlr9* specifically in B cells, B6 mice were prone to development of obesity with high fat diet, without an increase in food intake. The obesity was

accompanied by impaired glucose and insulin tolerance, together with high levels of fasting insulin in the circulation and fatty liver. Second, in the absence of *Tlr9* specifically in B cells, the KO mice had a reduced number of CD1d+CD5+ Breg cells and reduced IL-10-producing B cells in the peripheral lymphoid tissues but increased germinal center B cells. Moreover, B cell-specific deficiency in *Tlr9* also altered different subsets of T cells, including increased T memory cells, T follicular helper (T$_{FH}$) and T peripheral helper (T$_{PH}$) cells, most of which were IFNγ-producing cells. The proinflammatory immune cells contributed to the metabolic dysfunction in the KO mice and immunodeficient *Rag1-/-* mice following adoptive transfer of immune cells from KO mice. Third, B cell-specific deficiency in *Tlr9* altered the composition of the gut microbiota of the KO mice, which also contributed to metabolic dysregulation in both the KO mice themselves, as well as the germ-free recipients of fecal microbiota transfer from the KO donor mice. Co-abundant groups (CAG) 6 and 18 were particularly interesting in that CAG18, containing a high abundance of Lachnospiraceae, was significantly reduced in the *Tlr9fl/fl/Cd19Cre+/-* B6 (KO) mice, in addition to the altered Firmicutes to Bacteroidetes (F/B) ratio. Last, we identified a molecular link between *Tlr9* and *Irf4*, which regulated IL-10 production. Taken together, our in vivo, ex-vivo and in vitro results supported the notion that this pathway contributed to the features found in mice with B cell specific deficiency in *Tlr9*.

The absence of *Tlr9* in B cells resulted in reduced IL-10 at the transcription and protein levels in B cells. IL-10, an anti-inflammatory cytokine produced by a variety of immune cells, including T cells, B cells, and macrophages[65–67] plays a critical role in regulating immune responses by suppressing the activity of pro-inflammatory cytokines and promoting the differentiation of regulatory T cells[68–72]. IL-10 is also known to be particularly important in regulation of intestinal inflammation and in maintaining gut homeostasis[73–75]. It is not clear if the reduced IL-10 contributed to the alterations in the gut microbiota seen in our KO mice.

In addition to the changes we observed in different B cell subsets, *Tlr9* deficiency in B cells also altered T cell subsets, including T$_{FH}$ cells and T$_{PH}$ cells, both of which are closely related to the differentiation and function of B cells. *Tlr9* deficiency in B cells promoted IFN-γ- and IL-17a-producing CD4+ T cells. Furthermore, while splenic B cells from KO mice showed significantly impaired responses to adaptive stimuli, the T cell responses to adaptive stimuli were enhanced. For the innate pathways, B cells from the KO mice showed expected impaired responses to CpG, a ligand of TLR9. It is interesting that B cells from the KO mice had reduced responses to the TLR2 ligand, Pam3Csk4, but enhanced response to the TLR4 ligand, LPS. Interestingly, the KO mice had a higher abundance of Bacteroidetes, the major LPS-producing bacterial phylum. The observations that B cell-specific *Tlr9* deficiency reduced some B cell functions but enhanced LPS responses as well as enhancing pro-inflammatory T cell function suggest that TLR9 in B cells contributes more to the regulatory arm of the immune system. Inflammation significantly contributes to impaired insulin action, which promotes the progression of metabolic syndrome[76–78]. In our study, we found that proinflammatory responses were greater in immune cells from both lymphoid and WAT tissues in mice with *Tlr9*-deficient B cells compared to control mice, indicating that inflammation contributes to this process.

Increasing evidence suggests that gut microbiota play an important role in obesity and T2D[79–81]. In this study, we found that gut microbiota link *Tlr9* deletion in B cells to obesity and impaired metabolic responses. We incorporated ecological theory into the microbiota data analysis and, through network topology analysis of co-abundance groups (CAGs), we observed a dynamic imbalance between the decrease of CAG18, represented by Lachnospiraceae, and CAG6. These were significantly correlated with the phenotypic changes in the KO mice. The interaction between the immune system and the gut microbiota is complex and bidirectional, and the immune system can

have significant impact on the gut microbiota, and vice versa[82,83]. The B cell changes, including altered cytokine responses and intestinal antibody responses, could alter the homeostasis of the gut microbiome[84,85]. Our study showed that IgM was altered in both the small and large intestine, as well as in the serum. IgM is the first antibody produced during an immune response and is particularly effective at activating the complement system, a group of proteins that work together to destroy pathogens[86]. Recent research implied that IgM may play a role in regulating the gut microbiota and protecting against gut infections[86,87]. In addition, some studies have suggested that IgM may contribute to maintaining the diversity and stability of the gut microbiota[88]. It is conceivable that our IgM findings may be related to these functions.

In our study, the composition of gut microbiota was altered due to *Tlr9* deficiency in B cells, and these changes were accompanied by increased proinflammatory responses in the PP. Both immunological and metabolic changes in our KO mice could be transferred in FMT experiments using germ-free wild type B6 mice or germ-free *Rag1*⁻/⁻ B6 mice as recipients, which indicated a strong role for microbiota in modulating the immunological and metabolic changes. However, co-transfer of splenic immune cells from the KO mice, together with microbiota from control mice, did not fully recapitulate the phenotypes observed in the KO donor mice. We found that the metabolic and immunological phenotypes could only be fully recapitulated if the microbiota and the splenic immune cells were both from the KO donor mice.

At the molecular level, RNA sequencing analysis of purified B cells from *Tlr9fl/fl/Cd19Cre+/-* B6 and *Tlr9fl/fl/Cd19Cre-/-* B6 mice revealed that *Tlr9* via *Irf4* regulates *Il-10*. When we suppressed the IRF4 signaling in vitro using dexamethasone or siRNA, the phenotype of reduced IL-10 and increased IFNγ seen in *Tlr9fl/fl/Cd19Cre+/-* B6 mice was abolished. Our results provide evidence that the reduction in IL-10 secretion and increased IFNγ in *Tlr9fl/fl/Cd19Cre+/-* B6 mice is associated with IRF4. IRF4 plays an important role in the regulation of immune responses and is particularly involved in the transcriptional regulation of interferons[89,90]. IRF4 is primarily expressed in immune cells, and is important for the development and function of various immune cells, including T helper 2 (Th2) cells, T_{FH} cells, plasma cells, and memory B cells[91–93]. These are involved in the regulation of immune responses to infection and inflammation[91,92,94,95] and IRF4 and IL-10 interact to regulate immune responses[96]. Our in vitro experiments confirmed that B cells without *Tlr9*, or wild type B cells treated with TLR9 antagonist, led to significantly decreased expression of *Irf4* and *Il-10*, which suggests that *Tlr9* is required for upregulation of *Irf4* and *Il-10*.

Overall, our study using the mice with *Tlr9*-deficiency in B cells has delineated cross-talk among TLR9, adaptive immunity and gut microbiota, leading to an enhanced inflammatory milieu and susceptibility to obesity and associated metabolic changes. Obesity is a systemic metabolic disorder with diverse etiological factors and our study provides insight in the role of TLR9 in B cells in obesity. Thus, targeting TLR9 in B cells may have therapeutic potential for prevention and/or treatment of obesity and obesity-associated disorders.

## Methods

### Mice
C57BL/6 breeders were originally purchased from the Jackson Laboratory (#000664). *Tlr9fl/fl* C57BL/6 breeder mice were kindly provided by Dr. Mark Shlomchik[64] (University of Pittsburgh). *Cd19Cre+/+* C57BL/6 breeders were purchased from The Jackson Laboratory (RRID: IMSR_JAX:006785). *Tlr9fl/fl* and *Cd19Cre+/+* B6 mice were bred to generate *Tlr9fl/fl/Cd19Cre+/-*, which were then intercrossed to obtain B cell–specific *Tlr9*-deficient (*Tlr9fl/fl/Cd19Cre+/-*) and control (*Tlr9fl/fl/Cd19Cre-/-*) B6 mice. Germ-free (GF) C57BL/6 breeders[97] were kindly provided by Dr. Richard Flavell (Yale University) and bred in the Yale gnotobiotic mouse facility. *Rag1*⁻/⁻ GF C57BL/6 breeders (rederived to

GF at Yale from the #002216, the Jackson Laboratory) were kindly provided by Dr. Noah Palm (Yale University) and maintained in the Yale gnotobiotic mouse facility. Mice from different litters were randomly assigned to experimental groups. *Tlr9*-deficient (*Tlr9fl/fl/Cd19Cre+/-*) and control (*Tlr9fl/fl/Cd19Cre-/-*) B6 mice were housed in individually-ventilated cages in the SPF facility. GF mice were maintained in sterile isolators and verified monthly for GF status by PCR. All the mice used in this study were maintained under strict SPF or GF conditions with 71.7 F (22°C) temperature, 68 RH (relative humidity) and 12-h-dark/light cycle at the Yale School of Medicine, and all the protocols used for animal experimentation were approved by Yale Animal Care and Use Committee of Yale University.

### Diet
Mice were fed with normal diet (ND, 6 kcal% fat, Teklab Global, 2018SC) from weaning. Mice in the obesity study were fed with high fat diet (HFD, 60 kcal% fat, Research Diet, D12492) from 6- to 18-22 weeks old. Body weight was measured weekly. HFD food consumption was assessed by measuring the differences of the food weight between the food provided in the hopper of a clean cage and the remaining food in the hopper as well as the obvious food in the bedding during 24 h.

### Intraperitoneal glucose tolerance test (IPGTT)
Mice were allowed to fast for 12 h overnight, followed by intraperitoneal injection of glucose (1 g/kg for the mice with normal diet or 2 g/kg for the mice with high fat diet). Blood glucose concentration, from tail vein, was measured using a FreeStyle Precision Neo glucometer (Abbott) at 0 (prior to glucose injection), and 15, 30, 60 and 90 min after glucose injection.

### Insulin tolerance test (ITT)
Mice were fasted for 4 h, followed by intraperitoneal injection of insulin solution (0.75 U/kg, Humulin R, Lilly). Blood glucose concentration, from the tail vein, was measured using a FreeStyle Precision Neo glucometer (Abbott) at 0 (prior to insulin injection), and 15, 30, 60 and 90 min after insulin injection.

### Liver tissue collection and histology
Mice were euthanized by cervical dislocation under anesthesia according to American Association for Accreditation of Laboratory Animal Care (AAALAC) guideline and approved by Yale Institutional Animal Care & Use Committee (IACUC). The livers were dissected, weighed, and fixed in 10% paraformaldehyde for 24 h at 4 °C, prior to embedding with paraffin. Paraffin-embedded livers were sectioned at a thickness of 5 μm and stained with hematoxylin and eosin (H&E) to assess morphology.

### Immune cell extraction from spleen and lymphoid tissues
Bone marrow (BM), spleen (SPL), mesenteric lymph nodes (MLN) and Peyer's patches (PP), were dissected and collected into test tubes containing 3-5 ml sterile PBS. All the tissues were homogenized by gently grinding the tissue with the frosted end of glass slides followed by filtering through a nylon filter (150μm) to remove debris. Red blood cells in splenocytes were lysed by hypotonic shock.

### Immune cell extraction from adipose tissue
Ex-vivo epididymal fat tissue were cut into 1 mm small pieces and digested in collagenase I (1 mg/ml in completed RPMI, 10 ml/mice), followed by incubated at 37 °C with shaking (250 times/min) for 1 h. After stopping the digestion by adding 10 ml medium, the digested tissue was spun down at 800 g for 5 min. The cell pellet was resuspended in 2 ml ACK buffer and incubated at room temperature for 5 min. After adding 5 ml medium, the adipocytes were filtered through 100 μm nylon mesh followed by centrifugation (800 g) for 5 min. The resuspended cells were then analyzed using flow cytometry.

## Cell staining and flow cytometry

Single immune cell suspension (-10$^6$) from BM, SPL, MLNs, PP and white adipose tissue (WAT), were first incubated with Fc blocking antibody (2.4G2) for 15 min before staining for other surface markers. For intracellular cytokine staining, cells were incubated with 50 ng/ml phosphomolybdic acid (Sigma), 500 ng/ml of ionomycin (Sigma) and 1 µl of Golgi plug (BD Bioscience) at 37 °C for 4 h, followed by cell surface staining. After fixation with fixation buffer (Invitrogen) for 20 min (RT) and permeabilization (Invitrogen), cells were stained with antibodies against different cytokines. All the cells were also stained with Zombie Dye (Biolegend) to exclude the dead cells. The stained cells were analyzed on a BD LSR II flow cytometer. (Flow cytometry Antibody list: Supplement table1).

## Cell purification

Splenic B cells were purified using the EasySep Mouse B Cell Isolation Kit (STEMCELL Technologies) after depleting T cells by anti-Thy1(M5/49.4.1; BioXCell). Total spleen cells were incubated with 2.5µg/ml Thy1 and 1:20 complement (PEL-FREEZ BIOLOGICALS INC) at 37 °C, 5% CO$_2$ for 1 h. After washing with MoJo buffer (500 ml PBS + 10 ml FBS + 1 ml 0.5 mM EDTA) twice, the cells were purified using the B cell isolation kit according to the manufacturer's instructions. The purity of the cells was routinely >90%, confirmed by flow cytometry.

## Cell proliferation assay

Total spleen cells or purified B cells were stimulated with anti-CD40 (FGK4.5, BioXcell), with anti-IgM, or anti-CD3 (2C11 BioXcell) and anti-CD28 (37.51, BioXcell), or CpG (Invivogen), LPS (Invivogen) and Pam3Csk4 (Invivogen) at 37 °C, 5% CO2 for 3 days. The proliferative response was detected by $^3$H-thymidine incorporation and read on a β-counter. The cell proliferation was presented as counts per minute (cpm).

## Immunoglobulin (Ig) detection

Serum and gut contents were collected from 18-22-week-old HFD-treated *Tlr9$^{fl/fl}$/Cd19Cre$^{+/-}$* B6 and *Tlr9$^{fl/fl}$/Cd19Cre$^{-/-}$* B6 mice. The small intestine and colon were flushed with 5 ml sterile PBS and centrifuged at 800 $g$ for 5 min. Serum samples were centrifuged 1300 $g$ for 8 min. Immunoglobulins (IgM, IgG and IgA) were assessed in gut flush and serum samples by ELISA (Southern Biotech).

## Fecal microbial DNA extraction and bacterial 16 S rRNA gene sequencing

Fecal pellets were collected from each mouse before and after 12 to 16 weeks of HFD. Bacterial DNA extraction and 16 S rRNA gene sequencing were performed as described previously[98]. The results were quality-filtered using QIIME 2. The alpha diversity indices were compared with Kruskal-Wallis tests and the p values were adjusted for multiple comparison with Benjamini-Hochberg method[99]. The structure of the gut microbiota was compared using beta diversity with Bray-Curtis dissimilarity metric, visualized by principal-coordinate analysis (PCoA) plots. The statistical significance was assessed by permutational multivariate analysis of variance (PERMANOVA) via q2-diversity with 9,999 permutations and p values were adjusted with Benjamini-Hochberg correction method[99]. Taxonomic assignment for Amplicon Sequence Variants (ASVs) was performed via the q2-feature-classifier[100] using the Greengenes Expert Database[101].

## Microbial Co-abundant group (CAG) network analysis

CAG network analysis was used to identify the key responsive phylotypes. The correlations between the ASVs, which were shared among at least 20% of the samples, were calculated by using the SparCC method (bootstrap value, 100)[52]. Next, on the basis of SparCC correlation coefficient matrix, SparCC distance matrix (1-SparCC correlation coefficient) was calculated for the 125 shared ASVs, followed by

clustering into 32 CAGs through Ward's hierarchical clustering method and PERMANOVA with 999 permutations. A cluster tree was constructed and PERMANOVA was applied sequentially along the tree from top to bottom to identify the nodes with no significant difference as a single CAG (p > 0.001). The CAG network was then visualized in Gephi software.

## Fecal microbiota transfer (FMT) in germ-free C57BL/6 mice

Fecal pellets were collected from 18-22-week-old HFD-treated *Tlr9$^{fl/fl}$/Cd19Cre$^{+/-}$* B6 and *Tlr9$^{fl/fl}$/Cd19Cre$^{-/-}$* B6 mice and pooled. Pellets were homogenized and centrifuged at 800 $g$ for 5 min. Bacterial colony forming units (CFU) were measured by optical density (OD) with a known concentration of an *E. coli* strain as reference. Four-week-old germ-free (GF) C57BL/6 mice were colonized with fecal bacteria by oral gavage twice, 3 days apart (1 ×10$^8$ CFU/mouse) and kept in Sealed Positive Pressure (SPP) cages.

## Adoptive immune cell transfer and fecal microbiota transfer (FMT) in germ-free Rag1$^{-/-}$ C57BL/6 mice

Total splenocytes from 18-22-week-old HFD-treated *Tlr9$^{fl/fl}$/Cd19Cre$^{+/-}$* B6 and *Tlr9$^{fl/fl}$/Cd19Cre$^{-/-}$* B6 mice were adoptively transferred (i.v.) to 4-week-old germ-free *Rag1$^{-/-}$* mice (1 ×10$^7$/mouse), followed by fecal bacterial oral gavage as described above. In the experiment using SPF *Rag1$^{-/-}$* mice as recipients, the 4-week-old SPF *Rag1$^{-/-}$* mice were adoptively transferred (i.v.) with total splenic cells (1 ×10$^7$/mouse) only, without fecal bacterial transfer as SPF *Rag1$^{-/-}$* mice carry endogenous gut microbiota.

## B cell RNA sequencing and analysis

RNA was extracted from purified splenic B cells from 18-22-week-old HFD-treated *Tlr9$^{fl/fl}$/Cd19Cre$^{+/-}$* B6 and *Tlr9$^{fl/fl}$/Cd19Cre$^{-/-}$* B6 mice with RNeasy Mini Kit (Qiagen, Hilden, Germany) and quantified by Nano-Drop (ThermoFisher). RNA-sequencing (poly A) was performed at the Yale Center for Genome Analysis using NovaSeq with HiSeq paired-end, 100 bp. The raw fastq files were processed using fastp tool (version 0.20.0). With a default setting, sequencing reads with low-quality bases were trimmed or filtered. Alignment was performed for cleaned reads using STAR (version 2.7.9) and mouse reference genome (gencode version GRCm38.p6 with vM25 gene annotation). Expression quantification was performed for alignment results using featureCounts (version 2.0.0). As genes with low expression levels were most likely to be noise, they were excluded before downstream analysis, and we defined low expression filtering as expressed genes that should have ≥ 6 read counts in at least 20% of samples. The filtered read counts matrix was then normalized by transcripts per million (TPM) method. Detection of differentially expressed genes was performed using R package DESeq2 (version 1.30.1). The Benjamini-Hochberg procedure was used for multiple test correction, and FDR ≤ 0.05 as a threshold for detection of differentially expressed genes.

## Cell culture

Purified B cells (2 × 10$^6$) from 6-8-week-old wild type B6 mice were stimulated with 2 µg/ml TLR9 antagonist (ODN2088, Invivogen), ODN 2088 control (Invivogen) or 10µM dexamethasone at 37 °C, 5% CO$_2$ for 12 h.

## Quantitative Real-Time PCR

RNA was extracted from different tissues, using RNeasy Plus Mini Kit (Qiagen), prior to cDNA synthesis, using SuperScript III First-Strand Synthesis Kit with random hexamers (Invitrogen). Quantitative Real-Time PCR (qPCR) was performed with a qPCR cycler (iQ5; Bio-Rad). The relative gene expression was determined by the $2^{-\Delta\Delta CT}$ method and normalized with GAPDH reference gene. (Primer list: Supplementary table 2).

## siRNA delivery in vitro

IRF-4-specific siRNA (Horizon, USA) assay was performed following the Dharmacon™ Accell™ siRNA delivery protocol. Briefly, splenocytes ($5×10^5$/well) from KO and control mice were cultured in a 96-well plate in Accell siRNA Delivery Media (B-005000-500, Horizon, USA) for 48 h in the presence of 1 μM IRF4-specific siRNA at 37 °C, 5% $CO_2$. Controls included untreated cells and cells treated with non-targeting siRNA. The cells were harvested after 48 h and analyzed using flow cytometry post staining with appropriate immune markers.

## Statistical analysis

Statistical analysis was performed using GraphPad Prism software v9.0 (GraphPad Software, San Diego, CA, US). Differences between groups were analyzed with either a two-tailed Student's t test (if the data were normally distributed), a two-tailed Mann-Whitney test (if the data were not normally distributed), multiple t tests with Bonferroni correction, one-way ANOVA or a two-way ANOVA. $P < 0.05$ was considered significant.

## Reporting summary

Further information on research design is available in the Nature Portfolio Reporting Summary linked to this article.

## Data availability

All data generated and analyzed in this study are available from the corresponding author upon request. Source data are provided with this paper. The 16 S rRNA gene sequencing data generated in this study have been deposited in SRA database under accession code PRJNA1093493. Total B cell RNA sequencing data generated in this study have been deposited in NCBI's Gene Expression Omnibus under accession number GSE263396. Source data are provided with this paper.

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

## Acknowledgements

The authors thank Juan Carlos Roman, for taking care of the animals used in the study. This work was supported by The International Post-doctoral Exchange Fellowship Program of The Office of China Post-doctoral Council (No. 2021044) to X.Y., the National Institutes of Health (H.D. 097808, D.K. 126809, D.K. 130318, Diabetes Action Research and Education Foundation to L.W.); Diabetes Research Connection to Y.H. and L.W.; a JDRF Postdoctoral Research Fellowship (3-PDF-2016-197-A-N, 2016-2019) and a Medical Research Council Career Development Award (MR/T010525/1) to J.A.P., a Medical Research Council grant (MR/K021141/1 to F.S.W.). All the authors approved the final version of the manuscript and the submission.

## Author contributions

L.W. conceived the study, L.W., P.W., and X.Y. designed the study. P.W. researched most of the initial findings and X.Y. researched the critical experiments and data analysis. S.S., J.H., L.Y.Z., and J.P. researched some of the data. X.Y. and J.P. contributed to the data analysis of gut microbiota. J.G. and H.Z. analyzed the RNAseq results. P.W., X.Y., and L.W. wrote the manuscript. J.A.P. and F.S.W. revised the manuscript. QW was the consultant of the study.

## Competing interests

The authors declare no competing interests.
