## [Peer Review File · Nature Communications]

Tlr9 deficiency in B cells leads to obesity by promoting inflammation and gut dysbiosisREVIEWER COMMENTS

Reviewer #1 (Remarks to the Author):

This manuscript reports effects of TLR9 expressed by B cells on the obesity/metabolic disease. It shows that in the absence of TLR9 in B cells mice develop worse metabolic disease when fed HFD, and this phenotype can be transferred by the microbiota to healthy mice. B cells and T cells show multiple functional abnormalities in Bcell-Tlr9KO mice, and microbiota changes were detected. While this work is interesting and novel, there are several problems: inconsistent results between cell and microbiota transfer in different experiments, no effort to identify specific bacteria that could be responsible for the phenotype and other issues as described below. Finally, the pathway TLR9-IRF4-IL10 is not mechanistically supported since no IRF4 perturbation experiments were performed.

Specifically,

--the statement "the specific gut microbiota found in mice with B cell-Tlr9 deficiency was more important and sufficient to cause the observed alterations" is not correct. Both factors are required since KO bacteria did not transfer worse phenotype to mice with WT cells in fig 7G. To make it clear, all 4 groups from fig 7F-G should be depicted on the same graph that will show that only when BOTH Bcell-Tlr9KO cells and the KO microbiota were transferred, glucose tolerance was impaired. This is an interaction effect by 2-way ANOVA test that also detects if there is an effect of each factor separately.

--the absence of phenotype transfer by Bcell-Tlr9KO cells in fig. 7G contradicts the results of such transfer in fig. 7D and should be explained.

--a schematic for each experiment should be added to figs 6 and 7 alongside the results where different transfers were done because otherwise it's confusing. For example, experimental protocol shown in fig. S6 does not match the text of results, since no gavage of bacteria is shown. Timing is also incorrect based on the figure S6 (line 263 "observed them for 4 weeks").

--regarding microbiota analysis: only 5 mice were analyzed per group but many more were used in this study and should be included to find the robustness of results. This will increase the power of analysis and will likely detect more than just 4 taxa differentially abundant. Also, the bar chart is not informative and Table S1 is not clear. Overall, the analysis is very superficially presented and no analysis is described in the methods section.

--none of the mice after microbiota and cell transfer experiments were analyzed for microbiota composition. This analysis could allow to address the question of which specific bacteria are responsible for the phenotype transfer. At present, authors missed this opportunity.

--the pathway TLR9-IRF4-IL10 is not mechanistically supported since no IRF4 deletion experiments are shown. Is it known if stimulation of TLR9 in the absence of IRF4 results in lower IL10?

--regarding possible human relevance, is anything known about TLR9 and obesity in humans?

--what approaches do authors see for this statement that they make: "Targeting the TLR9 pathway in B cells and gut microbiota could offer novel preventive and therapeutic approaches for obesity"?

Reviewer #2 (Remarks to the Author):

The authors create mice with a B lineage restricted deficiency in TLR9 and study the effect of this deficiency on susceptibility to obesity and changes in the gut microbiota. The role of TLR9 in autoimmunity and B cell biology is well studied, and mice globally deficient in TLR9 have already

been examined for immunologic and metabolic changes during obesity (Hong et al. Obesity 2015 Aug 11; 23(11): 2199-2206). This new manuscript is adding to our understanding of the specific importance of B cell intrinsic expression of TLR9 during obesity, which is an important question. While this is an admirable goal, the authors have neglected to include highly relevant aspects of the previous literature, omitted an important control which leaves their conclusions in question, and missed an opportunity to provide a more concrete assessment of the changes they observe by providing numbers of cells in different organs rather than relative percentages. Including these important components is critical to the validity of the conclusions, as the data cannot be accurately assessed without them. Specific recommendations are below:

In the introduction and discussion, the authors describe their own work with global TLR9 deficient mice in a Type 1 Diabetes model, but do not mention a more relevant publication characterizing the immune populations and metabolic outcomes for global TLR9 deficient mice fed the same type of HFD they study here as part of an obesity study (Hong et al. Obesity 2015 Aug 11; 23(11): 2199-2206). It is especially surprising to see this paper omitted because many of the metabolic, immunologic, and inflammatory cytokine conclusions are similar.

In my opinion, the most significant problem is that the authors have utilized a CD19Cre murine line (JAX #006785) which we and many others have used for similar purposes. However, Jackson Labs provides key information about this strain in their online description, "The CD19-Cre knock-in/knock-out allele has a Cre recombinase gene inserted into the first coding exon of the CD19 antigen gene; both abolishing endogenous Cd19 gene function and placing cre expression under the control of the endogenous Cd19 promoter/enhancer elements." Accordingly, mice expressing one CD19Cre allele have CD19 haploinsufficiency, with reduced CD19 surface expression and function. CD19 is well known to regulate B cell signaling, and heterozygous expression compromises B cell responses. This caveat is discussed and relevant CD19Cre controls included in a paper from Mark Shlomchik, another Yale colleague, when his group considered the role of B cell intrinsic expression of TLR9 in a related autoimmune model of lupus (Tilstra et al, J Clin Invest. 2020 Jun 1; 130(6): 3172-3187, B cell-intrinsic TLR9 expression is protective in murine lupus).

Furthermore, another Yale colleague, Eric Meffre, published a critically relevant paper showing CD19 plays a specific role in directly regulating TLR9 responses in B cells (Morbach et al, J Allergy Clin Immunol. 2016 Mar; 137(3): 889-898.e6. CD19 controls TLR9 responses in human B cells). Given this caveat, the authors need to include CD19cre TLR9+/+ mice in their studies to control for the reduced expression of CD19 on the B cells. The need for this control is underscored by the data in Supplemental Fig 3 showing that immunologic changes observed in their obese HFD CD19Cre TLR9fl/fl animals were very similar or identical to significant changes observed in immune populations of NCD-fed lean CD19Cre TLR9fl/fl animals. Perhaps this phenotype is not even a consequence of the HFD/obesity, but a result of the immune changes imposed by the reduction in CD19 on the B cells.

The authors characterize a number of immune cell populations in spleen, LN, and Peyers Patches in Figures 2, 3, 4, 6 and Suppl3. The data is provided as a relative percentage, which can obscure changes in specific cell populations when many populations are fluctuating. This data needs to be provided as objective cell numbers per organ, not just relative proportions.

RESPONSE TO REVIEWERS' COMMENTS

Comments from Reviewer #1:

This manuscript reports effects of TLR9 expressed by B cells on the obesity/metabolic disease. It shows that in the absence of TLR9 in B cells mice develop worse metabolic disease when fed HFD, and this phenotype can be transferred by the microbiota to healthy mice. B cells and T cells show multiple functional abnormalities in B cell-Tlr9KO mice, and microbiota changes were detected.

While this work is interesting and novel, there are several problems: inconsistent results between cell and microbiota transfer in different experiments, no effort to identify specific bacteria that could be responsible for the phenotype and other issues as described below. Finally, the pathway TLR9-IRF4-IL10 is not mechanistically supported since no IRF4 perturbation experiments were performed.

We appreciate the time and effort dedicated to providing feedback on our work.

Regarding the fecal and/or cell transplantation experiments, we believe that the interplay between the gut microbiota, as an environmental factor, and the genetic background (specific knockout of *Tlr9* on B cells) collectively affects the host's metabolism. The genetic background of the recipient mice, regardless of whether the mice were immune-sufficient B6 mice or immune-deficient *Rag1*^{-/-} mice, were wild type in respect of TLR9, i.e., without TLR9 deficiency. Our experimental results showed that the TLR9-wild type germ-free recipient mice can partially replicate the immunological and metabolic phenotypes of the *Tlr9*^{fl/fl}/*Cd19Cre*^{+/-} donor mice, especially in the reduction of IL10 and the increase of IFN- γ (Figures in the main text of the revised manuscript are: **Figure 6 and Figure S11 in the revised manuscript**). Our data indicate that the genetic background also contributed to the perturbation of the gut microbiota seen in *Tlr9*^{fl/fl}/*Cd19 Cre*^{+/-} mice.

We have not identified specific bacterial strains, for the following reasons. First, the bacteria that affect the host metabolism are most likely to be an altered community rather than a limited number of specific bacterial strains. Obesity is the result of a broad biological dysregulation, i.e., dysbiosis, that involves the impact of bacteria on systemic chronic inflammation, food absorption, and basal metabolism, all of which may not be limited to a few single bacterial strains. Second, the gut microbiota analysis conducted in our study was by 16S rRNA-sequence that identifies species level but is unable to identify at bacterial strain level. Third, the potential causal role of the altered gut microbiota in metabolic abnormalities seen in our KO mice was experimentally validated, at least in part, through transplantation of fecal microbiota. In consideration of these factors, we treated the gut microbiota as a holistic community rather than focusing on a detailed analysis of individual strains. Having said that, we acknowledge the reviewer's comment, and therefore, we further analyzed our sequencing results using a different approach, categorizing 125 amplicon sequence variants (ASVs) into 22 Co-abundance Groups (CAGs) based on their co-varying abundances in the fecal DNA samples. Each CAG represents a group of bacteria showing similar changes in abundance.

We observed that two CAGs, CAG6 and CAG18, exhibited a negative correlation, exclusively, to the HFD condition and in CAG18, 8 out of 10 ASVs belong to the Lachnospiraceae family (**Fig. 5O** in the main text of the revised manuscript).

Interestingly, CAG18 showed a significant decrease in the knockout (KO) mice, whereas CAG6 exhibited a significant increase in the KO mice (**Fig. 5P and 5Q** in the main text of the revised manuscript). However, the taxonomic classification of the 10 ASVs in CAG6 was diverse.

Lachnospiraceae is a highly abundant bacterial family in the mammalian intestinal tract with a broad and complex taxonomic classification (DOI: 10.3390/microorganisms8040573). We speculate that the reduction of specific strains belonging to Lachnospiraceae may play a pivotal role in promoting the phenotype seen in the KO mice. However, we cannot rule out the possibility that the increase in

ASVs belonging to CAG6 may also contribute to the phenotype of the KO mice. The results generated from our additional analysis are presented in **Figure 5O-Q** in the revised main text and **Supplementary Figure 9**. We have described the CAG analysis in the results, methods, and discussion sections of the main text of the revision.

In relation to the reviewer's comment regarding IRF4. We have incorporated the results from the additional experiments that further support our mechanistic investigations.

Specifically,

--the statement “the specific gut microbiota found in mice with B cell-Tlr9 deficiency was more important and sufficient to cause the observed alterations” is not correct. Both factors are required since KO bacteria did not transfer worse phenotype to mice with WT cells in fig 7G. To make it clear, all 4 groups from fig 7F-G should be depicted on the same graph that will show that only when BOTH Bcell-Tlr9KO cells and the KO microbiota were transferred, glucose tolerance was impaired. This is an interaction effect by 2-way ANOVA test that also detects if there is an effect of each factor separately.

We agree and this also relates to our response described earlier, i.e., “both factors are required”. We did not pool the data from all the four groups into a single figure because the tests were not performed on the same day. Since the glucose tolerance test is a highly sensitive test, several factors can influence the baseline glucose levels. However, we have taken the reviewer's suggestion and merged the data from the four groups into a single figure for comparison (**Figure 7H** in the main text of the revised manuscript). We have also presented the area under the curve (AUC) of the tests (**Figure 7I** in the main text of the revised manuscript).

Germ-free *Rag1*^{-/-} recipient mice received both the immune cells and microbiota from the KO donors and exhibited the highest blood glucose levels among the four groups (**Figure 7H & I**). Interestingly, the SPF *Rag1*^{-/-} recipient mice (with endogenous gut

microbiota) that received only the immune cells from the KO donors also displayed higher blood glucose levels compared to those that received only the immune cells from the control donors (**Figure 7D-7G** in the main text of the revised manuscript). This suggests that the role of immune cells from the KO mice is also important in triggering abnormalities of glucose metabolism.

The data presented in **Figure 6A** in the original manuscript were the experiments in which the intestinal microbiota from the KO or control donor mice were transplanted into germ-free wild type B6 mice. Different from the germ-free *Rag1*^{-/-} B6 mice, germ-free wild type B6 mice have an intact immune system; thus, this set of experiments emphasized the contribution of the intestinal microbiota from the KO mice to the metabolic phenotype.

-the absence of phenotype transfer by Bcell-Tlr9KO cells in fig. 7G contradicts the results of such transfer in fig. 7D and should be explained.

We believe that the phenotypic contradictions observed in the original Figure 7G and 7D can be explained. First, the recipient mice were different. The previous Figure 7D (current **Figure 7D** and **Figure 7E** in the revision) were the data from the experiments using SPF *Rag1*^{-/-} mice that were transferred with the immune cells from B cell-Tlr9KO donors. The purpose of this experiment was to test the role of the immune cells of B cell-Tlr9KO mice in the hosts that have absence of immune cells but presence of gut microbiota. This compares with the original Figure 7G (current **Figure 7H** and **Figure 7I**) illustrating the data from **germ-free** *Rag1*^{-/-} recipient mice that were transferred with both the immune cells and gut bacteria from B cell-Tlr9KO donors. The purpose of this experiment was to test the role of the immune cells and gut bacteria from B cell-Tlr9KO donors as recipient mice that lack immune cells as well as gut bacteria. Moreover, we took a crisscross transfer approach. As *Rag1*^{-/-} mice are severely immunodeficient, the gut microbiota community in SPF *Rag1*^{-/-} mice would be very different from the gut microbiota community in B cell-Tlr9KO mice (immune-sufficient but B cell-Tlr9 deficient). Thus, the experimental design was to probe the impact of the immune cells from B cell-Tlr9KO mice on metabolism.

Additionally, diet was different in the duration of experimentation: the recipient mice in the original 7D (current **Figure 7D** and **Figure 7E**) were fed with high-fat diet and kept for 8 weeks (week 6 to week 14), while the recipient mice in the original 7G (current **Figure 7H** and **Figure 7I**) were fed with a normal diet and maintained for 4 weeks after immune cell and gut microbiota transfer. The reason that mice in 7G (Now is **Figure 7H** and **Figure 7I**) were not fed with high fat diet was because of a risk of contamination of germ-free mice by high fat diet, which requires special procedures to sterilize HFD that are not in our current standard operational protocol for germ-free mice. In summary, we have performed 3 sets of transfer experiments and the recipients were – 1) germ-free wild type B6 mice (immune-sufficient without microbiota); 2) SPF *Rag1*^{-/-} mice (immune-deficient with microbiota) and 3) germ-free *Rag1*^{-/-} mice (immune-deficient without microbiota). The donor materials used for the transfer experiments were – gut microbiota from KO and control donors for group (1); immune cells from KO and control mice for group (2); immune cells and gut microbiota from KO and control mice for group (3).

a schematic for each experiment should be added to figs 6 and 7 alongside the results where different transfers were done because otherwise it's confusing. For example, experimental protocol shown in fig. S6 does not match the text of results, since no gavage of bacteria is shown. Timing is also incorrect based on the figure S6 (line 263 “observed them for 4 weeks”).

We apologize for any confusion that might have caused. As described earlier, we performed three sets of transfer experiments. The schematic graph in the original **Fig.S6** was to illustrate the experiment using SPF *Rag1*^{-/-} B6 mice that were only transferred with immune cells intravenously as the SPF *Rag1*^{-/-} B6 mice have endogenous gut microbiota. These recipient mice were given HFD 2 weeks after the immune cell transfer and the experiment was terminated 8 weeks after HFD. The other set of experiments used germ-free *Rag1*^{-/-} B6 mice that were transferred with both immune cells (intravenously) and gut microbiota (orally).

To make it easier to understand for the reviewer and readers, we have included an additional schematic, as **Fig. S14** in the revised manuscript, depicting both immune cell transfer and microbiota gavage to germ-free *Rag1*^{-/-} B6 mice. Further, as described earlier, to avoid any possible bacterial contamination by the HFD to the recipient mice, which would affect the transplanted gut microbiota community, the recipient mice were not given HFD, but normal diet. The procedure to prepare germ-free HFD is different from that of germ-free normal diet. The recipient mice in this set of experiments were kept in special isocages with a specific handling protocol and observed over 4 weeks (when IPGTT was assessed) and terminated at week 5 after immune cells and gut microbiota transfer. We have redrawn all figures in the previous version of the manuscript and reorganized the layout of the figures to facilitate a clearer understanding of the narrative.

--regarding microbiota analysis: only 5 mice were analyzed per group but many more were used in this study and should be included to find the robustness of results. This will increase the power of analysis and will likely detect more than just 4 taxa differentially abundant. Also, the bar chart is not informative and Table S1 is not clear. Overall, the analysis is very superficially presented and no analysis is described in the methods section.

We apologize for lack of clarity, which may have caused some confusion about the data presented in Table S1. The numbers presented in the table on the 2nd row (Group) were not the number of mice in each group but rather the 5 different percentiles (corresponding to the percentiles on the 1st row) in each group. Anocom analysis is designated in 5 different percentiles (0, 25, 50, 75 and 100), and the numbers in the table represent each group of mice in each percentile, The number of the mice used for the microbiota analysis was, in fact, 11 - 13. We have used a different analytic strategy, and have removed the original Table S1 in the revised manuscript.

Taking the reviewer's comment into account, we have reanalyzed the data and provided a more in-depth interpretation of the amplicon sequencing data from an

ecological perspective. Please see **Figure 5O-Q** and **Supplementary Figure 9** in the revision.

Furthermore, we have revisited the alpha- and beta diversity analysis of the microbiota. We observed a significant increase in the Firmicutes/Bacteroidetes (F/B) ratio at the phylum level in the KO group, and the F/B ratio is a noticeable feature in obesity (DOI: 10.1038/nature05414). The figure of F/B ratio is presented as **Figure 5M-N** in the revised manuscript.

none of the mice after microbiota and cell transfer experiments were analyzed for microbiota composition. This analysis could allow to address the question of which specific bacteria are responsible for the phenotype transfer. At present, authors missed this opportunity.

We acknowledge the reviewer's concern. It is known that the germ-free recipients mostly retain the transferred microbiota community (PMID: 34621688, PMID: 30429801, PMID: 23912213). Therefore, we did not further confirm this. However, if the recipients are not germ-free, it is difficult to retain the same or similar microbiota community to that which was transferred, due to microbiome competition for survival. As such, scientists routinely treat the non-germfree recipients with high dose antibiotic cocktail to eliminate most of the endogenous gut microbiota before introducing new ones. It is also known that antibiotics affect the host metabolism, and our SPF recipient mice are severely immune-deficient. In a different project, we found that SPF *Rag1*^{-/-} mice were unhealthy and hunched after high dose antibiotic cocktail treatment, due to anorexia. Thus, the SPF *Rag1*^{-/-} recipient mice in the experiment were transferred with only immune cells from control or KO donors and the recipient mice retained their endogenous microbiota community. The results from this set of experiments showed that the immune cells from the KO donors altered the metabolism of the SPF *Rag1*^{-/-} hosts (**Fig.7A-7G** in the revised manuscript). However, we did not investigate how the immune cells from the KO donors affected the host metabolism and whether the immune cells from the KO donors altered the host endogenous gut microbiota community.

--the pathway TLR9-IRF4-IL10 is not mechanistically supported since no IRF4 deletion experiments are shown. Is it known if stimulation of TLR9 in the absence of IRF4 results in lower IL10?

To address the reviewer's comment, we performed additional experiments using dexamethasone that inhibits IRF4, and included the data in the revised manuscript (**Figure 8N-8P**).

Previous studies have shown that dexamethasone is an IRF4 inhibitor (DOI: 10.1016/j.isci.2023.108079 and DOI: 10.1016/j.jep.2018.12.016). We tested if dexamethasone affected IL-10-producing B cells and IFN- γ -producing T cells in an *in vitro* experiment. In this experiment, we used splenic immune cells from two control groups, the Cre control (Tlr9^{+/+} CD19Cre^{+/-}) and flox control (Tlr9^{fl/fl} CD19Cre^{-/-}). Supporting our original finding, B cells from KO mice had reduced expression of IRF4 in the absence of dexamethasone (3rd column in **Fig.8N**); however, dexamethasone highly inhibited IRF4 expression in B cells, across the board (column 4-6 in **Fig. 8N**). In line with our findings in the original manuscript, B cells from KO mice had significant reduction of IL10⁺ Breg cells (3rd column in **Fig.8O**) compared to the control mice. However, dexamethasone abolished the differences between KO and control mice (column 4-6 in **Fig.8O**). Also in line with our original findings that the KO mice showed highly increased IFN γ ⁺ T cells, especially IFN γ ⁺ CD4⁺ T cells (3rd column in **Fig. Fig.8P**), and dexamethasone abolished the differences (column 4-6 in **Fig.8P**). Although dexamethasone does not work solely on IRF4, our results suggest that the pro-inflammatory phenotype (reduction of IL10⁺ Breg cells and increasing IFN γ ⁺ CD4⁺ T cells) is, at least to some extent, mediated by IRF4.

regarding possible human relevance, is anything known about TLR9 and obesity in humans?

TLR9 has been demonstrated in some studies to influence lipid metabolism in humans. TLR9 can modulate the ratio of M1/M2 macrophages, thereby impacting hepatic lipid metabolism in humans (<https://doi.org/10.1042/CS20160838>). Study also

showed that TLR9 regulates inflammatory responses in both mouse and human adipose tissue and influences lipid metabolism levels (DOI: 10.1530/JOE-18-0326). It is noteworthy that the functions of immune cells are often compartmentalized in different tissues. Previous research has predominantly focused on the impact of TLR9 levels in the liver and white adipose tissue on lipid metabolism. However, the influence of TLR9 on glucose metabolism in B cells, which express abundant TLR9, remains unexplored. Our study addresses this knowledge gap using an animal model with specific deletion of TLR9 in B cells. While our research is not in humans, our findings, through different experimental approaches, provide pre-clinical evidence regarding the impact of TLR9 on host metabolism and its associated molecular mechanisms. We have addressed this in the Discussion section of the revision (Page16, Line 347-352).

--what approaches do authors see for this statement that they make: “Targeting the TLR9 pathway in B cells and gut microbiota could offer novel preventive and therapeutic approaches for obesity”?

Through our experiments, we have shown that TLR9 in B cells significantly influences the host’s metabolism as the hosts are prone to body weight gain, impaired glucose metabolism and altered gut microbiota in the absence of TLR9 in B cells. As mentioned earlier, our proof-of-concept pre-clinical study demonstrates that TLR9 in B cells is important in the homeostasis of the host metabolism and obesity associated gut microbiota. Thus, modulation of TLR9 in B cells may shed some light into the novel preventive and therapeutic intervention for obesity.

Comments from Reviewer #2:

The authors create mice with a B lineage restricted deficiency in TLR9 and study the effect of this deficiency on susceptibility to obesity and changes in the gut microbiota. The role of TLR9 in autoimmunity and B cell biology is well studied, and mice globally deficient in TLR9 have already been examined for

immunologic and metabolic changes during obesity (Hong et al. Obesity 2015 Aug 11; 23(11): 2199-2206).

This new manuscript is adding to our understanding of the specific importance of B cell intrinsic expression of TLR9 during obesity, which is an important question. While this is an admirable goal, the authors have neglected to include highly relevant aspects of the previous literature, omitted an important control which leaves their conclusions in question, and missed an opportunity to provide a more concrete assessment of the changes they observe by providing numbers of cells in different organs rather than relative percentages. Including these important components is critical to the validity of the conclusions, as the data cannot be accurately assessed without them.

We thank the reviewer for feedback on our work. We have now corrected our oversight and cited the paper in the revision. The reviewer's comment regarding the control group is well taken and ideally, both control groups (*Tlr9^{fl/fl}/Cd19Cre^{-/-}* and *Tlr9^{+/+}/Cd19Cre^{+/-}*) should be included. To explain, we bred *Tlr9^{fl/fl}/Cd19Cre^{+/-}* with *Tlr9^{fl/fl}/Cd19Cre^{+/-}* mice and the progeny were *Tlr9^{fl/fl}/Cd19Cre^{+/-}*, *Tlr9^{fl/fl}/Cd19Cre^{-/-}* and *Tlr9^{fl/fl}/Cd19Cre^{+/+}* without the *Tlr9^{+/+}/Cd19Cre^{+/-}* genotype. The mice with the genotype of *Tlr9^{fl/fl}/Cd19Cre^{+/-}* (KO) and *Tlr9^{fl/fl}/Cd19Cre^{-/-}* (control) were used for the study. If we had used the breeding strategy of *Tlr9^{fl/+}/Cd19Cre^{+/-}* with *Tlr9^{fl/+}/Cd19Cre^{+/-}*, we would have the progeny with 9 genotypes - *Tlr9^{fl/fl}/Cd19Cre^{+/-}*, *Tlr9^{fl/+}/Cd19Cre^{+/-}*, ***Tlr9^{+/+}/Cd19Cre^{+/-}***, *Tlr9^{fl/fl}/Cd19Cre^{+/+}*, *Tlr9^{fl/+}/Cd19Cre^{+/+}*, *Tlr9^{+/+}/Cd19Cre^{+/+}*, *Tlr9^{fl/fl}/Cd19Cre^{-/-}*, *Tlr9^{fl/+}/Cd19Cre^{-/-}* and *Tlr9^{+/+}/Cd19Cre^{-/-}*. Thus, the numbers of control mice, per the reviewer's comment, with genotype ***Tlr9^{+/+}/Cd19Cre^{+/-}*** would have been 1 in 9 and as our study used male mice, the possibility of obtaining male *Tlr9^{+/+}/Cd19Cre^{+/-}* male control mice would have been theoretically 1 in 18. Thus, practically obtaining sufficient male *Tlr9^{+/+}/Cd19Cre^{+/-}* control mice would not have been feasible, and would not be in accordance with the 3R policy (Replacement, Reduction and Refinement) for animal use. We also did not breed the *Cd19Cre^{+/-}* mice separately, as they would not have been littermates of the experimental mice (*Tlr9^{fl/fl}/Cd19Cre^{+/-}*) and the gut microbiota of these mice would

have been different from the experimental mice, related to maternal gut microbiota and not because of the presence of TLR9 in B cells (DOI: <https://doi.org/10.1016/j.chom.2023.01.018>, DOI: <https://doi.org/10.1016/j.chom.2020.06.009>). However, with limited mouse numbers, we have conducted some supplementary experiments, *in vivo* and *ex vivo*, to address the reviewer's concern and the results of those additional experiments were included in the **Supplementary Figure 1 and 2**.

Regarding the cell number, we fully acknowledge the reviewer's comment, we have included the number of the cells in the revision (**Supplementary Figures 5, 7, 8 and 11**).

Specific recommendations are below:

In the introduction and discussion, the authors describe their own work with global TLR9 deficient mice in a Type 1 Diabetes model, but do not mention a more relevant publication characterizing the immune populations and metabolic outcomes for global TLR9 deficient mice fed the same type of HFD they study here as part of an obesity study (Hong et al. Obesity 2015 Aug 11; 23(11): 2199-2206). It is especially surprising to see this paper omitted because many of the metabolic, immunologic, and inflammatory cytokine conclusions are similar.

We have cited Hong *et al*'s work in the revision and our findings further support the role of Tlr9 in metabolism. Hong *et al* 's work was conducted in mice with a complete knockout of TLR9. As one of the major types of immune cells with the high expression of Tlr9 (doi:10.1172/JCI89931; doi:10.1182/blood-2008-10-185421; doi:10.1002/eji.200636984; doi.org/10.1038/s41584-020-00544; doi:10.1038/nri1957), whether B cells located TLR9 has a specific function in regulating metabolism remains to be explored. Our study fills this gap. Further, in terms of the possible translation, it is unlikely to target TLR9 in all the tissue cells, it is conceivable to target TLR9 in only B cells.

In my opinion, the most significant problem is that the authors have utilized a CD19 Cre murine line (JAX #006785) which we and many others have used for similar purposes. However, Jackson Labs provides key information about this strain in their online description, “The CD19-Cre knock-in/knock-out allele has a Cre recombinase gene inserted into the first coding exon of the CD19 antigen gene; both abolishing endogenous Cd19 gene function and placing cre expression under the control of the endogenous Cd19 promoter/enhancer elements.”

Accordingly, mice expressing one CD19 Cre allele have CD19 haploinsufficiency, with reduced CD19 surface expression and function. CD19 is well known to regulate B cell signaling, and heterozygous expression compromises B cell responses. This caveat is discussed and relevant CD19Cre controls included in a paper from Mark Shlomchik, another Yale colleague, when his group considered the role of B cell intrinsic expression of TLR9 in a related autoimmune model of lupus (Tilstra et al, J Clin Invest. 2020 Jun 1; 130(6): 3172–3187, B cell-intrinsic TLR9 expression is protective in murine lupus).

Furthermore, another Yale colleague, Eric Meffre, published a critically relevant paper showing CD19 plays a specific role in directly regulating TLR9 responses in B cells (Morbach et al, J Allergy Clin Immunol. 2016 Mar; 137(3): 889–898.e6. CD19 controls TLR9 responses in human B cells). Given this caveat, the authors need to include CD19cre TLR9+/+ mice in their studies to control for the reduced expression of CD19 on the B cells. The need for this control is underscored by the data in Supplemental Fig 3 showing that immunologic changes observed in their obese HFD CD19Cre TLR9fl/fl animals were very similar or identical to significant changes observed in immune populations of NCD-fed lean CD19Cre TLR9fl/fl animals. Perhaps this phenotype is not even a consequence of the HFD/obesity, but a result of the immune changes imposed by the reduction in CD19 on the B cells.

In Mark Shlomchik’s study (Tilstra et al, J Clin Invest. 2020 Jun 1; 130(6): 3172–3187), the authors used Tlr9fl/fl cd19Cre^{-/-} mice as the controls (Fig.2 and 3), which was the same genotype control as we used in our study. In fact, the controls for the

other cell specific Tlr9 knockout mice (myeloid cell and dendritic cell) were all Tlr9^{fl/fl} without Cre mice in their study (Fig.4 & 5). The authors confirmed that CD19Cre mice (MRL/lpr background) were not different from the wild type MRL/lpr mice (Supplementary Fig.1A) and stated in their paper “CD19-Cre MRL/lpr mice were compared with MRL/lpr WT mice for disease outcomes. CD19-Cre MRL/lpr mice did not exhibit any significant differences in renal disease, dermatitis, lymph node weight, or spleen weight (Supplemental Figure 1A).” Dr. Eric Meffre and co-authors (Morbach et al, J Allergy Clin Immunol. 2016 Mar; 137(3): 889–898.e6) used human cells from patients with CD19 deficiency and human B cell lymphoma cell line (Ramos) with CD19 knockdown. The authors found the cross talk between TLR9 and CD19 in human B cells as B cells with CD19 deficiency (due to mutations in CD19 functional alleles) affected B cell activation induced by TLR9 stimulation. The authors also reported that B cells from patients with heterozygotic mutations expressed reduced activation measured by the expression of CD23/TACI and CD80/CD86 (Fig.1). Different from the human B cells shown in Dr. Eric Meffre’s JACI paper, the phenotype of heterozygotic CD19-Cre mice is comparable to the wild type mice. This was described at the Jackson Laboratory website (<https://www.jax.org/strain/006785>), in Dr. Mark Shlomchik’s study published in JCI and in our hands.

As discussed earlier, ideally, two control groups would have been used, but our strategy required practical consideration for generation littermate controls that is important for the investigation of gut microbiota in addition to the extremely low frequency of obtaining TLR9^{+/+} CD19cre^{+/-} mice.

However, we have conducted some supplementary experiments with the TLR9^{+/+} CD19cre^{+/-} control mice, *in vivo* and *ex vivo*, to directly address the reviewer’s concerns (**Supplementary Figure 1** and **Supplementary Figure 2**).

As predicted, we observed that anti-CD19 failed to bind CD19 on B cells (although anti-B220 binds B cells, data not shown) in all the lymphoid tissues tested including spleen, mesenteric lymph nodes (MLN), and Peyer's patches (PP) of *Cd19* cre^{+/+} mice. In contrast, there was no significant difference in CD19⁺ cells between *Tlr9*^{fl/fl} and

Cd19 cre^{+/-} mice. This indicated that the expression of CD19 in *CD19 cre^{+/-}* heterozygous mice was not noticeably affected, consistent with the description from Jackson Laboratory and Dr. Shlomchik's study. Thus, our findings using the KO mice (*Tlr9^{fl/fl} / Cd19Cre^{+/-}*) were therefore related to the specific deletion of *tlr9* in B cells (**Supplementary Figure 1**).

In addition, we compared the body weight changes in mice fed a high-fat diet among the three groups - *Ctrl_{flox}*, *Ctrl_{Cre}*, and KO mice. Supporting our original finding, we found that the KO mice gained significantly more body weight than that of the two control groups, whereas the body weight gain between the two control groups was comparable (**Supplementary Figure 2A**). Furthermore, we conducted a glucose tolerance test (GTT) on the three groups after a 6-week high-fat diet and there were no differences between the two control groups (**Supplementary Figure 2 B and C**). However, the blood glucose levels at 15- and 30-minute time points during IPGTT in the KO group mice exceeded the upper limit of the glucose meter's detection range and, therefore, we are unable to plot the data in the graph. The results from our additional experiments, *in vivo* and *ex vivo*, support our initial findings using *Tlr9^{fl/fl}* mice as controls.

The authors characterize a number of immune cell populations in spleen, LN, and Peyer's Patches in Figures 2, 3, 4, 6 and Suppl3. The data is provided as a relative percentage, which can obscure changes in specific cell populations when many populations are fluctuating. This data needs to be provided as objective cell numbers per organ, not just relative proportions.

We have showed all the data related to flow cytometric analysis in the manuscript with absolute values in the revision (**Supplementary Figures 5, 7, 8 and 11**).

Further, for the convenience of reviewers, editors and readers, we have made the following additional modifications to the manuscript:

- Enhanced the visual appearance of all figures, utilizing different color schemes for various experiments.
- We have included an analysis of the area under the curve (AUC) for the experimental data of IPGTT (Intraperitoneal Glucose Tolerance Test) and ITT (Insulin Tolerance Test).
- Renamed the mouse genotypes as follows: *Tlr9^{fl/fl}/Cd19Cre^{+/-}*, designated as KO group; *Tlr9^{fl/fl}/Cd19Cre^{-/-}*, designated as Ctr group.
- Simplified the categorization of different mouse groups in various experiments as follows: germ-free mice transplanted with fecal samples from the Ctr group, designated as GF_{Ctrl} group; germ-free mice transplanted with fecal samples from the KO group, designated as GF_{KO} group; immune cells stimulated with intestinal bacteria from mice in the Ctr group, designated as Bac_{Ctrl} group; immune cells stimulated with intestinal bacteria from mice in the KO group, designated as Bac_{KO} group; *Rag1^{-/-}* mice transplanted with splenocytes from the Ctr group, designated as Rag_{Ctrl} group; *Rag1^{-/-}* mice transplanted with splenocytes from the KO group, designated as Rag_{KO} group; for the experiments using germ-free *Rag1^{-/-}* mice, "Spl" represents the donor splenocytes, and "Bac" represents the donor intestinal bacteria.
- After a careful evaluation of the authors' contributions, to consider additional experiments performed for the revision, data analysis, as well as the contribution to the revision, we have revised the authorship order and added an additional author (Quan Wang). All the authors agreed with the authorship changes.

REVIEWER COMMENTS

Reviewer #1 (Remarks to the Author):

The authors answered several questions and now have a much better presentation of results/analyses but several concerns remain:

-the methods do not specify if the SparCC network construction was built from mice of the same genotype and on the diet - either KO or control mice? Or both together? If it was the latter, there might be the case of Simpson's paradox https://en.wikipedia.org/wiki/Simpson's_paradox and it should be checked. Depending on the result, the conclusion/discussion of results may change.

-please make sure to provide supplementary table corresponding to the network in fig. 5O and suppl. Fig. S9. The table should have average abundance of each of 125 ASVs and 32 CAGs in each mouse in each group, both diets.

-Authors should consider creating a heatmap for ALL differentially abundant ASVs/CAGs between genotypes on each diet instead of figs. 5P, 5Q. Also, two-way ANOVA (or PERMANOVA) should be used in this analysis.

-dexamethasone has a very dramatic effect regardless of genotype (hence differences disappear) and also it has many other inhibitory effects on immune pathways in B cells besides IRF4. It is surprising that the authors did not use simple siRNA anti-IRF4 in vitro in experiments in fig. 8N-P. Alternatively, B cells of IRF4 KO mice should be stimulated with TLR9-activating ligands and IL-10 production measured. Those cells might be available via collaboration with authors of this paper <https://pubmed.ncbi.nlm.nih.gov/16767092/> paper. Otherwise, the link TLR9IRF4 IL10 is rather weak and only suggestive.

-PLEASE merge experimental schemes in suppl figs with the corresponding results as much as possible on the SAME PAGE. It will be greatly appreciated by the readers (and reviewers).

Reviewer #2 (Remarks to the Author):

The authors have thoughtfully addressed all my previous concerns.

RESPONSE TO REVIEWERS' COMMENTS

Reviewer #1 (Remarks to the Author):

The authors answered several questions and now have a much better presentation of results/analyses but several concerns remain:

-the methods do not specify if the SparCC network construction was built from mice of the same genotype and on the diet - either KO or control mice? Or both together? If it was the latter, there might be the case of Simpson's paradox

https://en.wikipedia.org/wiki/Simpson's_paradox and it should be checked.

Depending on the result, the conclusion/discussion of results may change.

We acknowledge the importance of interpreting the findings in the context of the specific research question and data characteristics, and we exercise with caution when drawing conclusions, considering the influence of possible confounder(s).

In our study, we performed separate analyses for high-fat diet (HFD, Fig.5O) and normal diet (ND) groups (Fig. S8) as diet per se would have significant effect on gut microbiota. By analyzing the gut microbiota data from HFD and ND groups separately, we aimed at avoiding from the potential impact of Simpson's paradox on the experimental results. Additionally, we sought to investigate how the two different diets influence the gut microbiota in different genotypes (specific deletion of *Tlr9* on B cells or not) and whether there are differences in the interactions among gut microbiota as a complex community on the same diet.

We have revised the manuscript with clearer information about the mice and diet and addressed your comment in the updated revision (Page 10, Line 222-223).

-please make sure to provide supplementary table corresponding to the network in fig. 5O and suppl. Fig. S9. The table should have average abundance of each of 125 ASVs and 32 CAGs in each mouse in each group, both diets.

We have included the original data of Fig. 5O and Fig. S9 (new Fig.S8) (including CAGs abundance and CAGs to which ASVs belong) in the supplementary material.

These are in two supplementary Tables (Table 1 listed CAGs on HFD and Table 2 listed CAGs on ND).

-Authors should consider creating a heatmap for ALL differentially abundant ASVs/CAGs between genotypes on each diet instead of figs. 5P, 5Q. Also, two-way ANOVA (or PERMANOVA) should be used in this analysis.

We have created 2 heatmaps illustrating all differentially abundant CAGs between the two genotypes on each diet with PERMANOVA analysis (new Figure 5P and Figure S8B). However, regarding the previous Fig.5P and 5Q (new Figure 5Q and 5R), we think that it is better not to change because the spatial topology analysis shows that CAG6 and CAG18 have a mutually antagonistic characteristic on HFD, being CAG6 and CAG18 are highly abundant in the KO group and Ctr group respectively. We have revised the results section accordingly (Page11, Line 230-232).

-dexamethasone has a very dramatic effect regardless of genotype (hence differences disappear) and also it has many other inhibitory effects on immune pathways in B cells besides IRF4. It is surprising that the authors did not use simple siRNA anti-IRF4 in vitro in experiments in fig. 8N-P. Alternatively, B cells of IRF4 KO mice should be stimulated with TLR9-activating ligands and IL-10 production measured. Those cells might be available via collaboration with authors of this paper <https://pubmed.ncbi.nlm.nih.gov/16767092/> paper. Otherwise, the link TLR9->IRF4-> IL10 is rather weak and only suggestive.

We have performed the additional experiment with siRNA anti-IRF4 and the results are in line with the experiment results using dexamethasone. The results using siRNA anti-IRF4 are presented in the new Fig. 8Q. We think that the siRNA anti-IRF4 experimental results further support the notion of "TLR9-IRF4-IL10" link. We have added relevant content to the Results and Methods sections of the manuscript.

-PLEASE merge experimental schemes in suppl figs with the corresponding results as much as possible on the SAME PAGE. It will be greatly appreciated by the readers

(and reviewers).

We agree and have done accordingly.

REVIEWERS' COMMENTS

Reviewer #1 (Remarks to the Author):

The authors satisfactory addressed my comments. thanks!